# Nuclear spin diffusion in the central spin system of a GaAs/AlGaAs quantum dot

Peter Millington-Hotze [1], Santanu Manna[2,3], Saimon F. Covre da Silva [2], Armando Rastelli [2] & Evgeny A. Chekhovich [1] ✉

The spin diffusion concept provides a classical description of a purely quantum-mechanical evolution in inhomogeneously polarized many-body systems such as nuclear spin lattices. The central spin of a localized electron alters nuclear spin diffusion in a way that is still poorly understood. Here, spin diffusion in a single GaAs/AlGaAs quantum dot is witnessed in the most direct manner from oscillatory spin relaxation dynamics. Electron spin is found to accelerate nuclear spin relaxation, from which we conclude that the long-discussed concept of a Knight-field-gradient diffusion barrier does not apply to GaAs epitaxial quantum dots. Our experiments distinguish between non-diffusion relaxation and spin diffusion, allowing us to conclude that diffusion is accelerated by the central electron spin. Such acceleration is observed up to unexpectedly high magnetic fields – we propose electron spin-flip fluctuations as an explanation. Diffusion-limited nuclear spin lifetimes range between 1 and 10 s, which is sufficiently long for quantum information storage and processing.

Interacting many-body spin ensembles exhibit a variety of phenomena such as phase transitions[1,2] spin waves[3,4] and emergent thermodynamics[5,6]. Spin diffusion[7,8] is one of the earliest studied phenomena, where unitary quantum-mechanical evolution results in an irreversible dissipation of a localized spin polarization−a process that is well described by the classical diffusion model. Pure spin diffusion in homogeneous solids has been observed in a few notable examples[9,10]. However, most systems of interest are inhomogeneous by nature. In particular, magnetic (hyperfine) interaction with the central spin of a localized electron [Fig. 1a] causes shifts (known as the Knight shifts[11,12]) in the nuclear spin energy levels [Fig. 1b]. The resulting nuclear spin dynamics are complicated, as observed in a wide range of solid-state impurities[13–19] and semiconductor nanostructures[17,20–25]. Due to this complexity, it is still an open question whether the inhomogeneous Knight shifts accelerate[23,26,27] or suppress[16,25,27–30] spin diffusion between the nuclei. Resolving this dilemma is both of fundamental interest and practical importance for the recent proposals to use nuclear spins as quantum memories and registers[31–33], since

spin diffusion would set an ultimate limit to the longevity of any useful quantum state. Beyond semiconductor nanostructures, understanding of spin diffusion plays an important role in NMR signal enhancement and structural analysis of polymers[34,35], biomolecules[36–38], proteins[39], and pharmaceutical formulations[40].

Figure 1 sketches the central spin model where an electron can be trapped in a GaAs layer surrounded by the AlGaAs barriers, and for simplicity, spin-1/2 particles are used to describe the energy levels of the nuclei subject to the strong external magnetic field $B_z$. Any two nuclear spins $i$ and $j$ are coupled by the dipole-dipole interaction $\propto 2\hat{I}_{z,i}\hat{I}_{z,j} - (\hat{I}_{x,i}\hat{I}_{x,j} + \hat{I}_{y,i}\hat{I}_{y,j})$, where $\hat{I}_{x,i}, \hat{I}_{y,i}$ and $\hat{I}_{z,i}$ are the Cartesian components of the spin operator $\mathbf{I}_i$ of the $i$th nucleus. The $\propto (\hat{I}_{x,i}\hat{I}_{x,j} + \hat{I}_{y,i}\hat{I}_{y,j})$ term describes a flip-flop spin exchange process (curved arrows at $z = -1$ and 0 in Fig. 1b), responsible for the transfer of spin polarization in space, known as spin diffusion. The electric quadrupolar moments of the spin-3/2 nuclei make them sensitive to electric field gradients (EFGs), which can be induced by the GaAs/AlGaAs interface roughness ($z = 4.5$) or atomic-scale strains arising

[1]Department of Physics and Astronomy, University of Sheffield, Sheffield S3 7RH, United Kingdom. [2]Institute of Semiconductor and Solid State Physics, Johannes Kepler University Linz, Altenberger Str. 69, Linz 4040, Austria. [3]Present address: Department of Electrical Engineering, Indian Institute of Technology Delhi, New Delhi 110016, India. ✉e-mail: e.chekhovich@sheffield.ac.uk

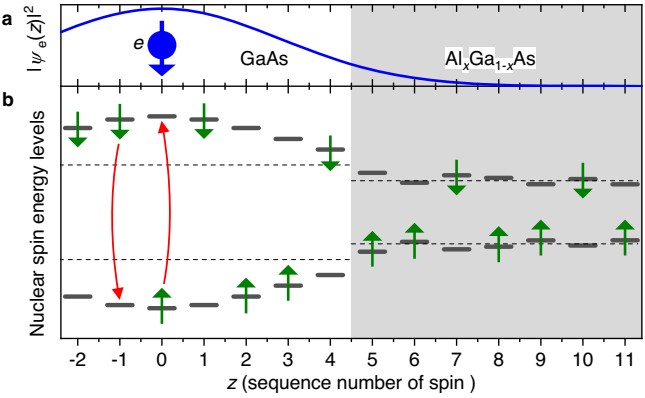

**Fig. 1 | Schematic of a central spin model.** The sketch is for the one-dimensional case, along the growth axis $z$ of a GaAs/AlGaAs structure. **a** Wavefunction density $|\psi_e|^2$ of an electron ($e$, ball with arrow) localized in GaAs. **b** Energy levels of the nuclei, that are depicted for simplicity as spins 1/2, and can occupy states with +1/2 and −1/2 spin projections (up and down arrows). Dashed lines show the bulk nuclear spin energies dominated by the external magnetic field $B_z$. These bulk energies are generally different in GaAs ($z \leq 4$) and AlGaAs ($z \geq 5$) due to the difference in chemical shifts and homogeneous strain. The energies of the individual nuclei are further shifted by the electron Knight field (mainly in GaAs) and by the atomic-scale strain disorder in the AlGaAs alloy. Magnetic-dipole interaction between the nuclei can result in spin exchange via a flip-flop process, sketched by the curved arrows for nuclei at $z = −1$ and $z = 0$ as an example. If energy mismatch is larger than the nuclear spin level homogenous broadening, for example for nuclei at $z = 4$ and $z = 5$, the spin exchange becomes prohibited, suppressing nuclear spin diffusion.

from random positioning of the aluminium atoms[41,42] in AlGaAs ($z \geq 5$). These quadrupolar effects lead to mismatches in the energy splittings of adjacent nuclei, which in turn impede the nuclear spin diffusion.

When an electron is added, its spin **s** couples to the nuclear spin ensemble via hyperfine interaction:

$$\hat{H}_{hf} = \sum_j A_j(\hat{s}_x\hat{I}_{x,j} + \hat{s}_y\hat{I}_{y,j} + \hat{s}_z\hat{I}_{z,j}), \qquad (1)$$

where the summation goes over all nuclei $j$, and the coupling energies $A_j$ are proportional to the electron density $|\psi_e(\mathbf{r}_j)|^2$ at the nuclear sites $\mathbf{r}_j$. There are two competing effects of the hyperfine interaction. On the one hand, through the term $\hat{s}_z\hat{I}_z$, the electron spin can produce a further diffusion barrier[16,25,27–30], at the points of strong Knight shift gradient ($z = 3$ in Fig. 1a). On the other hand, the electron spin can mediate spin flip-flops between two distant nuclei with similar energy splitting (e.g., $z = −2$ and $z = 2$), potentially opening a new channel for spin diffusion, especially at low magnetic fields[23,26,27]. Both of these effects have been known for decades, and both were claimed to be dominant in different previous studies, often without giving consideration to the other alternative. The main purpose of this study is to settle this dilemma through systematic experimental work.

Here, we examine electron-controlled nuclear spin diffusion in high-quality epitaxial GaAs/AlGaAs quantum dots (QDs), which emerged recently as an excellent platform for quantum light emitters[43–45] as well as spin qubits[33,46] and quantum memories[32]. Crucially, we design experiments where nuclear spin dynamics are examined either in the absence or in the presence of the electron central spin, but under an otherwise identical initial nuclear spin state. In this way, we distinguish with high accuracy the effects specific to the electron spin. This allows us to demonstrate that no observable Knight-field-gradient diffusion barrier is formed. Instead, the nuclear–nuclear interactions, mediated by the electron spin, accelerate nuclear spin diffusion up to unexpectedly high magnetic fields—we attribute this to the impact of the electron spin flips. Our results answer a long-standing question in spin physics, and provide practical guidelines for the

design and optimization of quantum dot electron-nuclear spin qubits and quantum memories.

## Results

### Sample and experimental techniques

The studied heterostructure is grown by in situ etching of nanoholes[47,48] in the AlGaAs surface [Fig. 2a, b], which are then infilled with GaAs to form the QDs. The structure is processed into a $p-i-n$ diode [Fig. 2c] where an external bias $V_{Gate}$ is applied to charge QDs deterministically with individual electrons (See details in Supplementary Note 1). In this way, it is possible to study nuclear spin dynamics in an empty ($0e$) or single electron ($1e$) state. A static magnetic field $B_z$ is applied along the growth axis $z$ (Faraday geometry) and the sample is kept at a liquid helium temperature of 4.3 K. We use a confocal microscopy configuration where QD photoluminescence (PL) is excited and collected through an aspheric lens with a focal distance of 1.45 mm and numerical aperture of 0.58. The collected PL is dispersed in a two-stage grating spectrometer, and recorded with a charge-coupled device (CCD) camera.

The change in the PL spectral splitting $\Delta E_{PL}$ of a negatively charged trion $X^-$ [see Fig. 2d] is the hyperfine shift $E_{hf}$, which gives a measure of an average spin polarization degree of the $\approx 10^5$ QD nuclei[12]. The hyperfine shifts (also known as Overhauser shifts) arise from the $\hat{s}_z\hat{I}_z$ term of the hyperfine interaction Hamiltonian [Eq. (1)]. Large nonequilibrium nuclear spin polarization is generated on demand by exciting the QD with a circularly polarized pump laser[12], which repeatedly injects spin-polarized electrons into a QD, and causes nuclear spin polarization build up via electron-nuclear spin flip-flops described by the $\hat{s}_x\hat{I}_x + \hat{s}_y\hat{I}_y$ part of Eq. (1). A small copper wire coil is placed near the sample to produce radiofrequency (RF) oscillating magnetic field perpendicular to the static magnetic field. Application of the RF field allows for the energy spectrum of the nuclear spins to be probed via nuclear magnetic resonance (NMR). Moreover, the RF field can be used to depolarize the nuclear spins on demand. Further details can be found in Supplementary Note 2, including sample growth details, PL spectra, characterization of QD charge state control, and additional results at an elevated temperature of 15.2 K.

### Nuclear spin system of a GaAs quantum dot

Figure 2e shows NMR spectra of $^{75}$As in a single GaAs QD, measured using the "inverse NMR" technique with an optical Pump-RF-Probe cycle shown in the top inset. For an empty QD (open symbols), an NMR triplet is observed[49], corresponding to the three magnetic-dipole transitions between the four Zeeman-split states $I_z = \{−3/2, −1/2, +1/2, +3/2\}$ of a spin-3/2 nucleus (left inset). The central resolution-limited peak originates from the $−1/2 \leftrightarrow +1/2$ NMR transition that is weakly affected by strain. The two satellite transition peaks $\pm 1/2 \leftrightarrow \pm 3/2$ are split from the central transition peak by the strain-induced EFGs. The average splitting $\nu_Q \approx 24$ kHz between the triplet components corresponds to an average elastic strain of $\approx 2.6 \times 10^{-4}$ (refs. 50,51). The satellite transitions are inhomogeneously broadened, with non-zero NMR amplitudes detected approximately in a range of $\nu_Q \in [10, 50]$ kHz, indicating that elastic strain varies within the nanoscale volume of the QD. The $^{69}$Ga and $^{71}$Ga nuclear spins are also affected by the strain, but the quadrupolar shifts $\nu_Q$ are smaller by a factor of $\approx 2$ and $\approx 3$, respectively[50–52].

When a single electron occupies the QD, it induces inhomogeneous Knight shifts that exceed the quadrupolar shifts, leading to a broadened NMR peak [solid symbols in Fig. 2e]. From the NMR peak width, the Knight frequency shifts, characterizing the typical coupling strength between the electron spin and an individual nuclear spin, are estimated to be $A_j/(2h) \approx 50$ kHz, where $h$ is Planck's constant.

These NMR characterization results indicate a complex interplay of dipolar, quadrupolar, and hyperfine interactions governing the nuclear spin dynamics, which we now investigate experimentally.

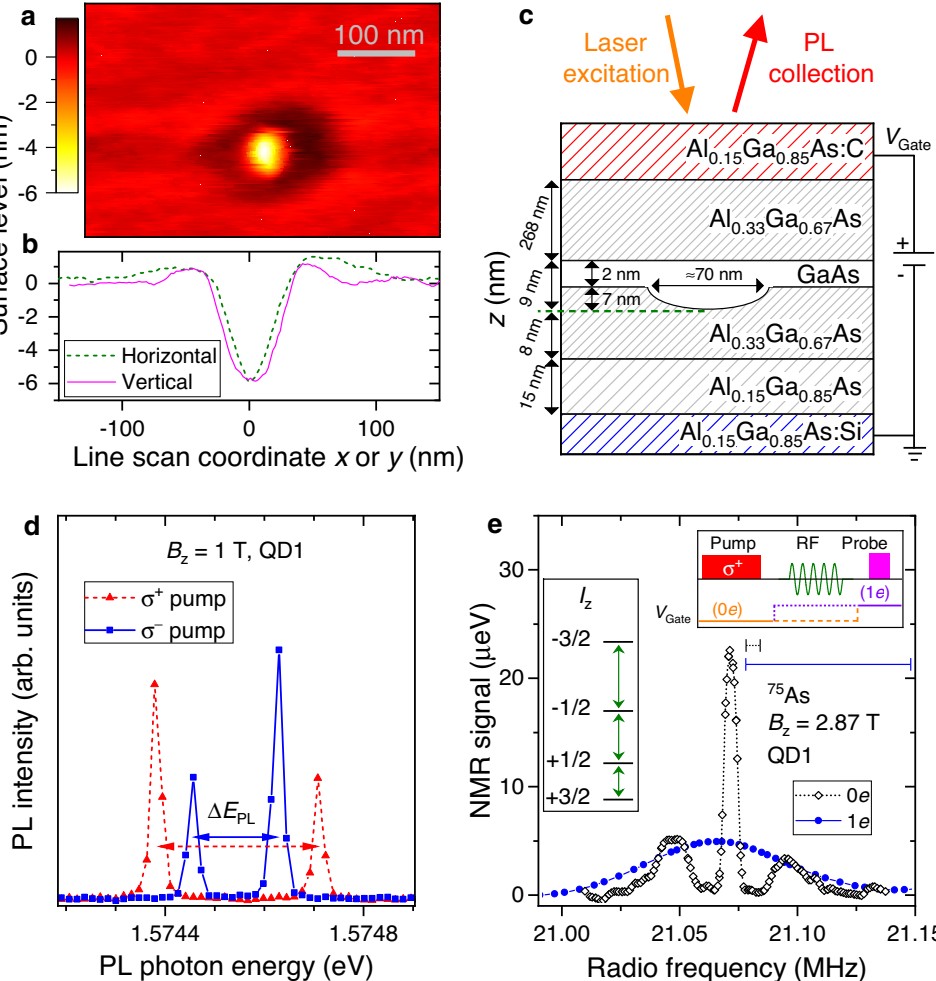

**Fig. 2 | Optically active epitaxial GaAs/AlGaAs quantum dots. a** Atomic force microscopy (AFM) profile of the AlGaAs surface after nanohole etching. **b** Surface level profiles taken along the horizontal and vertical lines through the center of the nanohole in (**a**). **c** Schematic (not to scale) of the sample structure. GaAs QDs are formed by infilling of the in situ etched nanoholes in the bottom $Al_{0.33}Ga_{0.67}As$ barrier. The bottom (top) $Al_{0.15}Ga_{0.85}As$ layer is $n$ ($p$) type doped to form a $p - i - n$ diode structure. External gate bias $V_{Gate}$ is applied for deterministic QD charging with electrons. **d** Photoluminescence (PL) spectra of a negatively charged trion $X^-$, following either $\sigma^+$ (triangles) or $\sigma^-$ (squares) optical pumping, which induces nuclear spin polarization. This polarization manifests in hyperfine shifts $E_{hf}$ of the Zeeman doublet spectral splitting $\Delta E_{PL}$. **e** Optically detected NMR of the $^{75}$As spin-3/2 nuclei measured in a single QD. Strain-induced quadrupolar shifts of the nuclear spin-3/2 levels (left inset) give rise to an NMR triplet with splitting $\nu_Q \approx 24$ kHz, observed in an empty QD (0$e$, diamonds). Charging the QD with a single electron (1$e$, circles) induces inhomogeneous Knight shifts observed as NMR spectral broadening. The measurement is conducted using the "inverse NMR" signal amplification technique[68], with spectral resolution shown by the horizontal bars (smaller for 0$e$ and larger for 1$e$). The measurement Pump-RF-Probe cycle is shown in the top inset. The bias $V_{Gate}$ is tuned to 0$e$ charge state for the optical pumping of the nuclear spins and to 1$e$ state for their optical probing. The radio-frequency (RF) pulse is applied in the dark under either 0$e$ or 1$e$ bias.

## Observation of nuclear spin diffusion in a GaAs quantum dot

While nuclear spin diffusion is a well-known phenomenon, its direct observation is rarely possible[9,10]. Thus we start with an experiment that reveals spin diffusion in a QD structure in a most convincing manner. The measurement cycle [see timing diagram in Fig. 3a] starts with a long $\sigma^+$ polarized optical pump. It creates a negative nuclear polarization degree $P_N$ that diffuses out of the QD into the surrounding material. The resulting spatial profile of $P_N(z)$ is depicted in the leftmost sketch in Fig. 3b. Then, a much shorter $\sigma^-$ pump is applied. This second pump is too short for diffusion to take place, so a positive $P_N$ is localized only in a QD, while the surrounding remains negatively polarized (second sketch in Fig. 3b). This two-stage pumping (similar to "hole burning" implemented previously in shallow donors[15]) is followed by a dark time $T_{Dark}$. Finally, the remaining polarization within the QD volume (i.e., around $z = 0$) is probed through an optically detected hyperfine shift $E_{hf}$. The measured dependence $E_{hf}(T_{Dark})$ is plotted in Fig. 3b and shows non-monotonic spin dynamics. A sign-reversal occurs at $T_{Dark} \approx 10$ s when the negative $P_N$, induced by the first

pump and stored in the surrounding barriers, refluxes back into the QD. This diffusion reflux peaks around $T_{Dark} \approx 100$ s where $E_{hf}$ reaches its minimum. At even longer $T_{Dark}$ nuclear spin polarization decays monotonically towards $E_{hf} \approx 0$.

We point out that the thermal-equilibrium hyperfine shifts are very small $|E_{hf}| \lesssim 0.15$ µeV, so that, any non-zero $E_{hf}$ can only arise from dynamical nuclear spin polarization. The non-diffusion nuclear spin relaxation (NSR) mechanisms, such as direct spin-lattice coupling and hyperfine interaction with electrons, can only lead to monotonic decay of $E_{hf}$ towards ≈0. Spatial transfer of polarization is the only mechanism that can produce non-monotonic free evolution and sign-reversal of $E_{hf}$. Another way to describe the diffusion reflux experiment is to note that switching between $\sigma^+$ and $\sigma^-$ essentially corresponds to time-oscillating nuclear spin pumping, which creates a wave-like initial spatial profile. In the subsequent free evolution, this spatial polarization wave is converted back into temporal oscillations of nuclear polarization at the QD site. To our knowledge, such oscillating spin relaxation

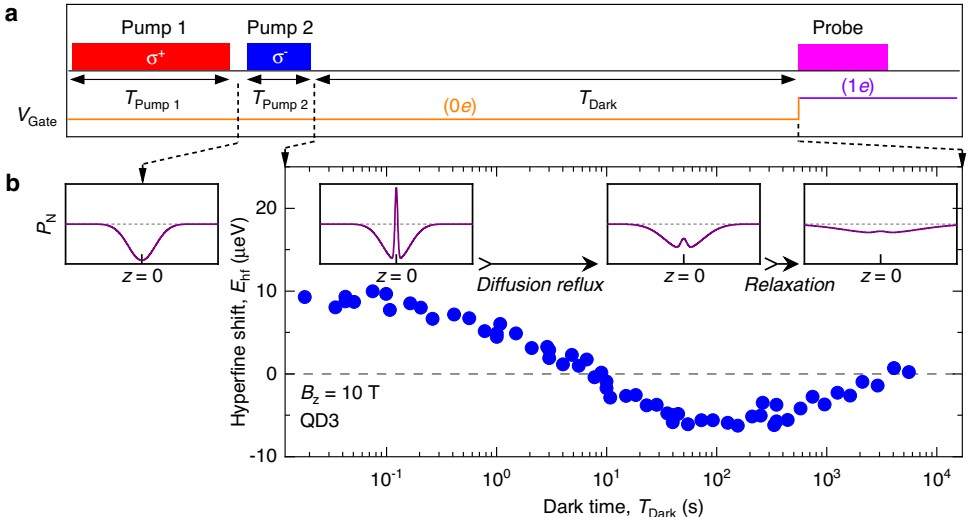

**Fig. 3 | Oscillatory nuclear spin relaxation due to spin diffusion reflux. a** The measurement cycle starts with a $\sigma^+$ polarized laser pump pulse that generates negative nuclear spin polarization with a steady-state hyperfine shift $E_{hf} \approx -36$ µeV. As demonstrated below, this first pump is long enough ($T_{Pump\,1} = 90$ s) for spin polarization induced in the QD to diffuse into the surrounding AlGaAs barriers. This is followed by the second $\sigma^-$ pump, which is kept short ($T_{Pump\,2} = 0.1$ s) in order to create an inverted positive ($E_{hf} \approx +10$ µeV) nuclear spin polarization localized to the QD volume only. Nuclear spin polarization prepared in this way is then allowed to evolve freely over the variable dark time $T_{Dark}$ while keeping the QD empty of any charges (0$e$) through a large reverse gate bias $V_{Gate}$. Finally, a short probe laser pulse induces photoluminescence (PL). The hyperfine shifts $E_{hf}$ detected in PL spectra provide a measure of the average polarization of $\approx 10^5$ QD nuclear spins, weighted by the QD electron density $|\psi_e|^2$. **b** Dark time dependence of the hyperfine shift $E_{hf}$ measured in an individual QD3 reveals non-monotonic (oscillating) nuclear spin relaxation. This indicates a diffusion reflux where spin polarization induced by the first pump pulse and stored in the surrounding barriers, diffuses back into the QD. The four insets sketch the spatial profiles $P_N(z)$ of the nuclear spin polarization degree following the two pump pulses and at the different stages of nuclear spin relaxation. The QD is located at $z = 0$.

gives by far the most direct evidence of nuclear spin diffusion between an individual QD and its surrounding.

## Nuclear spin relaxation in a GaAs quantum dot

We now proceed to quantitative NSR measurements with a timing diagram shown in Fig. 4a. First, any remnant nuclear spin polarization is erased by saturating the $^{75}$As, $^{69}$Ga, and $^{71}$Ga NMR resonances in the entire heterostructure[53]. This is followed by a single variable-duration ($T_{Pump}$) optical pumping pulse[15,17,20,22]. In order to localize the nuclear spin polarization to the QD nanoscale volume, we choose the pump photon energy to be below the AlGaAs barrier bandgap. After the pump laser is turned off, the gate bias $V_{Gate}$ is set to a desired level for a dark time $T_{Dark}$—this way evolution under 0$e$ or 1$e$ QD charge state is studied for nominally identical initial nuclear spin polarizations. Finally, $E_{hf}$ is measured optically, which provides nuclear spin polarization averaged over all nuclei of the QD. The relative isotope contributions to $E_{hf}$ arising from $^{75}$As, $^{69}$Ga, and $^{71}$Ga are $\approx$49, 28, and 23%, respectively[54].

Figure 4b shows the average QD nuclear spin polarization as a function of the pump-probe delay $T_{Dark}$ during which the sample is kept in the dark. The decay is non-exponential, thus we characterize the NSR timescale $T_{1,N}$ by the half-life time over which the QD hyperfine shift $E_{hf}$ decays to 1/2 of its initial value. The NSR rate is then defined as $\Gamma_N = 1/T_{1,N}$. When the pumping time $T_{Pump}$ is increased, $T_{1,N}$ notably increases, as can be seen in Fig. 4c, d. Such dependence of $T_{1,N}$ on $T_{Pump}$ is observed both in empty (0$e$) and charged (1$e$) QD states, and in a wide range of magnetic fields.

## Relaxation dominated by nuclear spin diffusion

In order to explain the results of Fig. 4, we note that nuclear spin dipole-dipole interactions conserve the nuclear spin polarization for any magnetic field exceeding the dipolar local field, typically $\lesssim$1 mT. Therefore, at a high magnetic field the decay of nuclear spin polarization can proceed via two routes: either via spin-conserving diffusion to the surrounding nuclei, or spin transfer to external degrees of

freedom, including quadrupolar coupling to lattice vibrations[16,55] or a hyperfine interaction with a charge spin[16,56–58] that is in turn coupled to the lattice or other spins. Spin diffusion can only take place if the spatial profile of the initial nuclear spin polarization is inhomogeneous, as exemplified in the reflux experiment in Fig. 3. By contrast, direct spin-lattice and hyperfine interactions have no explicit dependence on the spin polarization spatial profile. Optical pumping time $T_{Pump}$ that is short compared to spin diffusion timescales creates nuclear spin polarization localized to the QD volume[15,17,20,22]. Therefore, observation of short $T_{1,N}$ at short $T_{Pump}$ is a clear indicator that spin diffusion is the dominant NSR mechanism in the studied QDs. Conversely, if the pumping duration $T_{Pump}$ is long, there is enough time for nuclear polarization to diffuse from the QD into the surrounding AlGaAs barriers, suppressing any subsequent spin diffusion out of the QD and increasing $T_{1,N}$, as observed in Fig. 4c, d.

In order to complement our experimental investigation we model the spatiotemporal evolution of the nuclear spin polarization degree $P_N(t, z)$ by solving numerically the one-dimensional spin diffusion equation

$$\frac{\partial P_N(t,z)}{\partial t} = D(t)\frac{\partial^2 P_N(t,z)}{\partial z^2} + w(t)|\psi_e(z)|^2(P_{N,0} - P_N(t,z)), \quad (2)$$

where the last term describes optical nuclear spin pumping with a rate proportional to electron density $|\psi_e(z)|^2$ and the time-dependent factor $w(t)$ equal to 0 or $w_0$ when optical pumping is off or on, respectively. Correspondingly, the spin diffusion coefficient $D(t)$ takes two discrete values $D_{Dark}$ or $D_{Pump}$ when optical pumping is off or on, respectively. $P_{N,0}$ is a steady-state nuclear spin polarization degree that optical pumping would generate in the absence of spin diffusion. Eq. (2) is solved numerically and the parameters such as $D_{Dark}^{(ne)}$, $w_0(B_z)$, $D_{Pump}(B_z)$ are varied to achieve the best fit to the entire experimental datasets of $E_{hf}(T_{Pump}, T_{Dark})$ measured at $B_z = 0.39$ and 9.82 T for empty ($n = 0$) and charged ($n = 1$) QD states. The best fit calculated dynamics are shown by the lines in Fig. 4b and capture well the main features of the

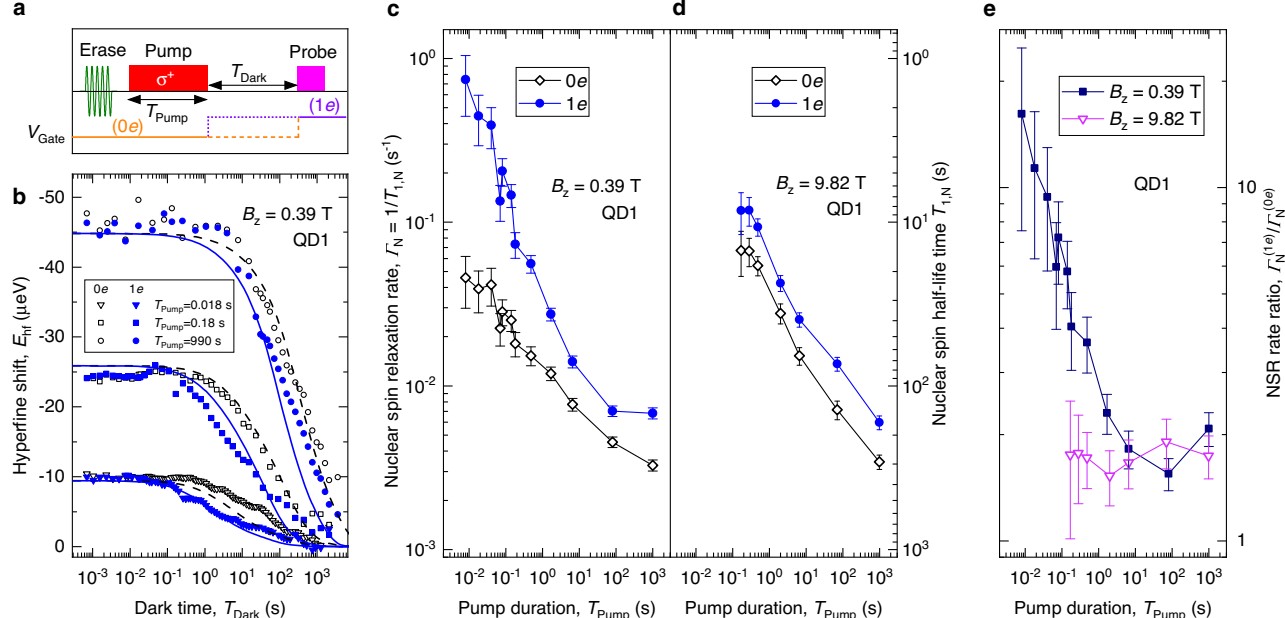

**Fig. 4 | Nuclear spin relaxation in a quantum dot. a** NSR measurement cycle starting with a radiofrequency erase pulse, followed by circularly polarized ($\sigma^+$) optical pumping and then optical probing of the QD nuclear spin polarization after dark evolution delay $T_{Dark}$ (see details in Supplementary Note 2). The sample gate bias $V_{Gate}$ is varied throughout the experiment cycle. During $T_{Dark}$ the bias can be set to achieve either 0e (dashed line) or 1e (dotted line) QD charge state. **b** Dark time dependence of the hyperfine shift $E_{hf}$. The nuclear spin decay is measured (symbols) at $B_z = 0.39$ T for different pumping times $T_{Pump}$ while keeping the QD empty (0e, open symbols) or charged with one electron (1e, solid symbols) during the dark time. Lines show the numerical solution of the spin diffusion equation [Eq. (2)]. **c** Fitted QD nuclear spin half-lifetimes $T_{1,N}$ (right scale) and the corresponding NSR rates $\Gamma_N = 1/T_{1,N}$ (left scale) measured as a function of the pumping time $T_{Pump}$ at magnetic field $B_z = 0.39$ T. **d** same as (**c**) for $B_z = 9.82$ T. **e** Ratio $\Gamma_N^{(1e)}/\Gamma_N^{(0e)}$ of the NSR rates in 1e and 0e charge states as a function of $T_{Pump}$ measured at $B_z = 0.39$ T (squares) and $B_z = 9.82$ T (triangles). All results are for the same individual dot QD1. Error bars are 95% confidence intervals.

experimentally measured nuclear spin decay, confirming the validity of the spin diffusion picture. The one-dimensional character of diffusion, occurring predominantly along the sample growth $z$ direction, is justified by the large ratio of the QD diameter $\approx 70$ nm to QD height < 9 nm, and is further verified by modeling two-dimensional spin diffusion (see Supplementary Note 4).

**Effect of central spin on nuclear spin diffusion**

Dividing the typical Knight shift of $\approx 50$ kHz by half the QD thickness (4.5 nm) we calculate the gradient and roughly estimate the Knight shift difference of $\approx 4.4$ kHz for the two nearest-neighbor spins of the same isotope separated by $a_0/\sqrt{2}$ (here, $a_0 = 0.565$ nm is the lattice constant). The energy corresponding to such a difference significantly exceeds the energy that can be exchanged with the nuclear dipole-dipole reservoir for a spin flip-flop to happen[25] (the dipole-dipole energy is on the order of $\approx h/T_{2,N}$, where $T_{2,N} \in [1, 5]$ ms is the nuclear spin-echo coherence time[33,59]). The flip-flops would then be limited to the few nuclear spin pairs whose vector differences are nearly orthogonal to the Knight field gradient. Therefore, one may naively expect a Knight-field-gradient barrier to form and suppress spin diffusion in an electron-charged QD. By contrast, Fig. 4c, d show that in an experiment the NSR is faster when the QD is occupied by a single electron (1e, solid symbols) for all studied $T_{Pump}$, demonstrating that no significant Knight-field-gradient barrier is formed. However, in order to quantify the effect of the central spin on nuclear spin diffusion we must distinguish it from other non-diffusion NSR mechanisms introduced by the electron spin. To this end, we examine the magnetic field dependence shown in Fig. 5.

First, we examine a case where long optical pumping is used to suppress spin diffusion, thus highlighting the non-diffusion NSR mechanisms. Figure 5a shows the experimental dependence $\Gamma_N(B_z)$ for long $T_{Pump} = 990$ s. The results indicate that in an empty QD (0e) spin diffusion is still the dominant NSR mechanisms at $T_{Pump} = 990$ s.

Indeed, the observed rates $\Gamma_N^{(0e)} \in [1 \times 10^{-3}, 6 \times 10^{-3}]\,s^{-1}$ are considerably higher than those found in bulk crystal experiments[55], where spin diffusion is negligible, resulting in relaxation rates as low as $\Gamma_N \approx 6 \times 10^{-5}$ in semi-insulating GaAs[16]. The electron-induced (1e) rates under long pumping $\Gamma_N^{(1e)} \in [4 \times 10^{-3}, 2 \times 10^{-2}]\,s^{-1}$ are nearly independent of $B_z$, and exceed the 0e rates by no more than a factor of $\Gamma_N^{(1e)}/\Gamma_N^{(0e)} < 4$ [squares in Fig. 5c]. Such a small effect of the electron is explained by the small strain of the GaAs/AlGaAs structures, which reduces the efficiency of the non-diffusion NSR mechanisms related to phonon and electron cotunneling. This is in stark contrast to the large magnetic field-induced variation $\Gamma_N^{(1e)} \in [5 \times 10^{-4}, 1 \times 10^1]\,s^{-1}$ in Stranski–Krastanov self-assembled InGaAs QDs[58], where phonon and cotunneling non-diffusion mechanisms dominate, both enabled by the noncollinear hyperfine interaction[56,58], arising in turn from the large strain-induced nuclear quadrupolar shifts.

In the case of long optical pumping, the NSR rates are nearly constant, exhibiting only a small irregular dependence on the magnetic field [Fig. 5a]. The long-pumping absolute NSR rates are also consistent across different individual QDs, as demonstrated in Fig. 5b. The main reason for the residual scatter in Fig. 5a, b is the dot-to-dot variation and magnetic field dependence of the QD optical absorption spectrum. As a result, the same optical pump power and wavelength lead to a different nuclear spin pumping rate, which affects the initial spatial profile of the nuclear spin polarization and the subsequent spin diffusion dynamics. Other uncontrollable parameters may include the charge state of the nearby impurities. While the absolute NSR rates $\Gamma_N^{(1e)}$ and $\Gamma_N^{(0e)}$ are subject to uncontrollable effects, their ratio $\Gamma_N^{(1e)}/\Gamma_N^{(0e)}$ is a robust quantity. This is exemplified in Fig. 4e, where at high magnetic field $B_z = 9.82$ T (triangles) the rate ratio is seen to be constant, even though the absolute rates depend strongly on $T_{Pump}$ (Fig. 4d). At low $B_z = 0.39$ T there is a significant dependence of $\Gamma_N^{(1e)}/\Gamma_N^{(0e)}$ on the pumping time $T_{Pump}$ (squares in Fig. 4e). Therefore, we use the

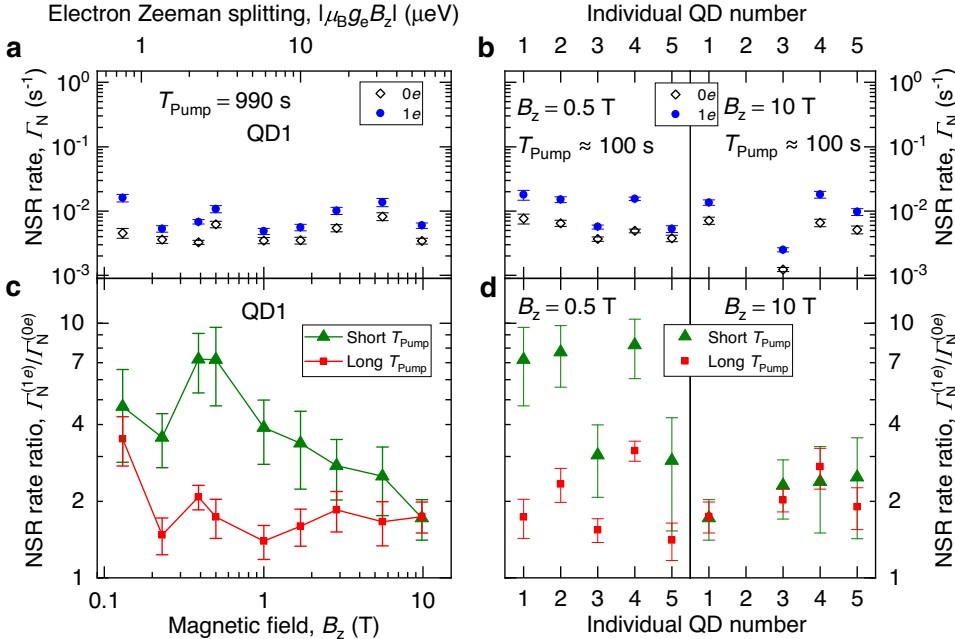

**Fig. 5 | Magnetic field dependence of spin diffusion. a** Nuclear spin relaxation (NSR) rate $\Gamma_N$ as a function of $B_z$ measured in $0e$ (open symbols) and $1e$ (solid symbols) QD charge states upon long pumping $T_{Pump} = 990$ s. The top horizontal axis shows the electron Zeeman splitting at zero nuclear spin polarization. **b** $\Gamma_N$ for different individual dots QD1-QD5 at $B_z = 0.5$ T (left) and $B_z = 10$ T (right) measured with a long pumping time $T_{Pump} = 70$–$100$ s. **c** Ratio $\Gamma_N^{(1e)}/\Gamma_N^{(0e)}$ of the NSR rates in $0e$

and $1e$ charge states as a function of $B_z$ measured under long pumping $T_{Pump} = 990$ s (squares) and short pumping $T_{Pump} \in [0.08, 0.6]$ s (triangles). **d** Ratio $\Gamma_N^{(1e)}/\Gamma_N^{(0e)}$ for QD1-QD5 at $B_z = 0.5$ T (left) and $B_z = 10$ T (right). Short pumping is $T_{Pump} \in [0.08, 0.6]$ s, whereas long-pumping time is $T_{Pump} = 990$ s for QD1-QD2 and $T_{Pump} = 100$ s for the remaining QDs. Error bars are 95% confidence intervals.

$\Gamma_N^{(1e)}/\Gamma_N^{(0e)}$ ratio to gauge the electron spin's effect on NSR, including its impact on spin diffusion.

In order to discriminate the diffusion-related effect of the QD electron spin, in addition to the long-pumping measurements discussed above [squares in Fig. 5c], we choose for each magnetic field a short pumping time, typically $T_{Pump} \in [0.08, 0.6]$ s, that yields initial QD nuclear spin polarization at $\approx 1/2$ of the steady-state long-pumping polarization. The resulting short-pumping ratio $\Gamma_N^{(1e)}/\Gamma_N^{(0e)}$ is shown by the triangles in Fig. 5c. The ratio $\Gamma_N^{(1e)}/\Gamma_N^{(0e)}$ combines all the electron-induced effects. However, the significant excess of the short-pumping ratio $\Gamma_N^{(1e)}/\Gamma_N^{(0e)}$ [triangles in Fig. 5c] over the long-pumping ratio $\Gamma_N^{(1e)}/\Gamma_N^{(0e)}$ [squares in Fig. 5c] is ascribed to spin diffusion alone, discriminating it from any non-diffusion mechanisms introduced by the electron spin. The electron spin-induced acceleration of the nuclear spin diffusion is seen to be particularly pronounced at low magnetic fields $B_z \lesssim 0.5$ T, consistent with the influence of the electron-mediated nuclear–nuclear spin interaction[23,26,27]. Such pairwise indirect interaction of nuclei $j$ and $k$ is derived from the second-order perturbation expansion of Eq. (1):

$$\mathcal{H}_{hf,j,k}^{ind} \propto \frac{A_j A_k}{\Delta E_e} \hat{s}_z \hat{I}_j^{(+)} \hat{I}_k^{(-)}, \qquad (3)$$

where $\hat{I}_j^{(\pm)} = \hat{I}_{x,j} \pm i\hat{I}_{y,j}$ and $\Delta E_e = \mu_B g_e B_z + E_{hf}$ is the electron spin splitting due to both the Zeeman effect and the nuclear spin-induced hyperfine shift $E_{hf}$. In our experiments, both contributions are negative, so that any nuclear spin pumping increases $|\Delta E_e|$. The rate of the indirect nuclear–nuclear spin flip-flops scales as $\propto \Delta E_e^{-2}$. Consequently, the resulting acceleration of nuclear spin diffusion in gate-defined GaAs QDs was previously found to be limited to the low fields $B < 0.02$–$0.75$ T (refs. 23,27,60). By contrast, Fig. 5c shows that such acceleration persists at unexpectedly high magnetic fields, well above $B_z \gtrsim 2$ T. The short- and long-pumping $\Gamma_N^{(1e)}/\Gamma_N^{(0e)}$ ratios converge only at the maximum field $B_z = 9.82$ T.

Figure 5d shows $\Gamma_N^{(1e)}/\Gamma_N^{(0e)}$ for five different QDs in the same sample. Since it is too time-consuming to measure full dependence, a fixed $T_{Pump}$ was chosen for each QD, which inevitably leads to variation in the actual $\Gamma_N^{(1e)}/\Gamma_N^{(0e)}$ ratios. However, for all QDs, we observe an excess of the short-pumping ratios over the long-pumping ratios at $B_z = 0.5$ T, which becomes negligible at $B_z = 10$ T. This confirms that electron-induced acceleration of nuclear spin diffusion is a systematic effect.

**Acceleration of spin diffusion at high magnetic fields**

We now examine why the electron-induced acceleration of nuclear spin diffusion is observed at high magnetic fields. The electron $g$-factor in the studied epitaxial QDs is $g_e \approx -0.1$ (see Supplementary Note 2), much smaller than $g_e \approx -0.4$ in the gate-defined QDs. Moreover, the number of nuclei is an order of magnitude smaller in our epitaxial QDs. These factors result in a smaller $|\Delta E_e|$ and larger $A_j$, respectively, which should lead to a stronger hyperfine-mediated coupling in the studied QDs (Eq. (3)). However, this difference does not explain the magnetic field dependence. At high field $B_z = 9.82$ T the electron spin Zeeman splitting is $|\Delta E_e| \approx 58$ μeV. At low field $B_z = 0.39$ T we take into account both the Zeeman splitting $\approx -2.3$ μeV and the time-averaged hyperfine shift $E_{hf} \approx -2.5$ μeV (half of the initial $E_{hf} \approx -5$ μeV under the shortest used $T_{Pump} \approx 8$ ms) to estimate $|\Delta E_e| \approx 4.8$ μeV. This suggests a factor of $(58/4.8)^2 \approx 150$ reductions in the hyperfine-mediated rates. However, the measured short-pumping NSR rate for QD1 reduces only by a factor of $\approx 6$ from $\Gamma_N^{(1e)} \approx 0.74$ s$^{-1}$ at low field [Fig. 4c] to $\Gamma_N^{(1e)} \approx 0.12$ s$^{-1}$ at high field [Fig. 4d]. Prompted by these observations, we point out that Eq. (3) treats the central electron spin as isolated, while in a real system, the electron is coupled to external environments such as phonons and other charges.

A fluctuating electron spin can accelerate nuclear spin diffusion, provided there is a frequency component in the time-dependent Knight field that equals the energy mismatch of a pair of nuclei[61,62]. This contribution has been considered for deep impurities[14], and, as we now

discuss, should also be taken into account in the context of III-V semiconductor nanostructures. Electron spin flips are always present due to phonons and cotunneling coupling to the electron Fermi reservoir of the $n$-doped layer[58,63,64]. It is worth noting that the acceleration of spin diffusion discussed in this paragraph is distinct from the non-diffusion NSR mechanisms, where the phonon bath and Fermi reservoir act as a sink for the nuclear spin momentum, carried through the electron spin. Preliminary studies of the relaxation dynamics are conducted using single-shot readout of the electron spin via nuclei[65] (see Supplementary Note 5, details to be reported elsewhere). Electron spin lifetimes are found to be $T_{1,e} \approx 7$ ms at $B_z = 2$ T, reducing to $T_{1,e} \approx 0.5$ ms at $B_z = 7$ T. The electron flips are dominated by phonons and occur as abrupt jumps (telegraph process). Hence the Knight field should have a significant spectral density around the [1, 10] kHz range, matching the typical differences in the nuclear spin energies found from NMR spectra of Fig. 2e. Thus we speculate that the electron spin flips contribute to the acceleration of nuclear spin diffusion in the studied GaAs QDs, especially at high magnetic fields. In other words, the widely used model of hyperfine-mediate nuclear–nuclear interactions [Eq. (3)] considers only the zero-frequency component, whereas our data suggest that the entire electron spin fluctuation spectrum must be included. Our explanation is supported by numerical modeling (Supplementary Note 4), which yields a significant increase in the nuclear spin diffusion coefficients under optical pumping $D_{Pump} \gg D_{Dark}$ where electron spin flips are accelerated[66]. Future work may address this phenomenon through the measurement of nuclear spin diffusion under simultaneous flipping of the central electron spin with microwave pulses.

**Comparison with previous results on nuclear spin diffusion**

In order to understand what controls the rate of spin diffusion we first make a comparison with Stranski–Krastanov InGaAs/GaAs and InP/GaInP self-assembled QDs, where quadrupolar shifts are so large (MHz range[67,68]) that all nuclear spins are essentially isolated from each other, eliminating spin diffusion and resulting in very long nuclear spin lifetimes $T_{1,N}^{(0e)} > 10^4$ s in empty (0$e$) QDs[26,29,56,58,69,70]. Even in the presence of the electron spin (1$e$) the nuclear spin diffusion takes place only inside the QD[26,56], without diffusion into the surrounding material.

In the lattice-matched GaAs QDs, the strain-induced effects are smaller but not negligible, characterized by quadrupolar shifts $v_Q$ ranging approximately between 10 and 50 kHz within the QD, as revealed by NMR spectra in Fig. 2e. Nuclei in $I_z = \pm 1/2$ and $|I_z| > 1/2$ states must be considered separately. The central transition between the $I_z = -1/2$ and $+1/2$ spin states is affected only by the second-order quadrupolar shifts, which scale as $\propto v_Q^2/v_L$ and are within a few kHz for the studied range of nuclear spin Larmor frequencies $v_L \in [1, 130]$ MHz. These second-order quadrupolar shifts are comparable to the homogeneous nuclear spin linewidth $\propto 1/T_{2,N}$, and therefore spin diffusion in GaAs/AlGaAs QDs is expected to be nearly unimpeded for the nuclei in the $I_z = \pm 1/2$ states. By contrast, the $I_z = \pm 3/2$ spin states experience first-order quadrupolar shifts $v_Q$, which are tens of kHz, significantly exceeding the homogeneous NMR linewidths in the studied GaAs QDs. The resulting dynamics of the $I_z = \pm 3/2$ nuclei are therefore sensitive to nanoscale inhomogeneity of the strain-induced $v_Q$. Such inhomogeneity is expected to be most pronounced for $^{75}$As in the AlGaAs barriers, where random positioning of Ga and Al atoms produces unit-cell-scale strains[41,68]. From the NSR experiments [Fig. 4b], we observe that nuclear spin polarization relaxes to zero, even in an empty QD (0$e$). This can only happen if spin diffusion is unimpeded not only for the $I_z = \pm 1/2$ states, but also for the $I_z = \pm 3/2$ states that are subject to the larger first-order quadrupolar shifts. Our interpretation is that strain in the studied GaAs/AlGaAs QDs is a smooth function of spatial coordinates: for nearly each QD nucleus it is possible to find some neighboring nuclei with a strain variation small enough to form a chain that conducts spin diffusion out of the GaAs QD into the AlGaAs barriers.

Similarly fast NSR was observed previously in neutral QDs formed by monolayer fluctuations in GaAs/AlGaAs quantum wells[22]. However, the opposite scenario was realized in QDs with nanoholes etched in pure GaAs[49] where nuclear spin polarization in an empty QD (0$e$) was preserved for over $T_{1,N} > 5000$ s, suggesting that some of the nuclei were frozen in the $I_z = \pm 3/2$ states, akin to quadrupolar blockade of spin diffusion in Stranski–Krastanov self-assembled QDs. This contrast is rather remarkable since the average strain, characterized by the average $v_Q \approx 20$–30 kHz, is very similar for QDs grown in nanoholes etched in AlGaAs (studied here) and in GaAs (ref. 49). This comparison suggests that bare nuclear spin dynamics (without the electron) are sensitive to QD morphology down to the atomic scale, and could be affected by factors such as QD shape, as well GaAs/AlGaAs interface roughness and intermixing[71–73]. One possible contributing factor is the QD growth temperature, which was 610° C in the structures used here, considerably higher than 520° C in the structures studied previously[48,49]. Further work would be required to elucidate the role of all the underlying growth parameters. Conversely, NSR can be a sensitive probe of the QD internal structure.

We now quantify the spin diffusion process and compare our results to the earlier studies in GaAs-based structures. The best fit of the experimental NSR dynamics [lines in Fig. 4b] yields $D_{Dark}^{(0e)} = 2.2_{-0.5}^{+0.7}$ nm$^2$ s$^{-1}$ (95% confidence interval) for the diffusion coefficient in an empty QD and in the absence of optical excitation, in reasonable agreement with $D = 1.0 \pm 0.15$ nm$^2$ s$^{-1}$ measured previously for spin diffusion between two GaAs quantum wells across an Al$_{0.35}$Ga$_{0.65}$As barrier[41]. This is approximately an order of magnitude smaller than the first-principle estimate[74–76] of $D_{Dark}^{(0e)} \approx 19$ nm$^2$ s$^{-1}$ for bulk GaAs (see Supplementary Note 3) and the $D = 15.0 \pm 7$ nm$^2$ s$^{-1}$ value measured in pure AlAs[77]. The reduced diffusion in the AlGaAs alloy can be explained by the quadrupolar disorder, arising from the random positioning of the aluminium atoms[41]. Charging of the QD with a single electron accelerates spin diffusion: we find $D_{Dark}^{(1e)}(9.82\,\text{T}) = 4.7_{-1.0}^{+1.2}$ nm$^2$ s$^{-1}$, which increases to $D_{Dark}^{(1e)}(0.39\,\text{T}) = 7.7 \pm 1.9$ nm$^2$ s$^{-1}$ at low magnetic fields where hyperfine-mediated nuclear–nuclear spin exchange is enhanced in accordance with Eq. (3). While the spin diffusion Eq. (2) gives an overall good description of the experimental data in Fig. 4b, some residual deviation is also apparent. The imperfect fit could be linked to a range of simplifications, such as ignoring the spatial variations of the nuclear–nuclear couplings and the dependence of the electron spin splitting $\Delta E_e$ on the instantaneous nuclear spin polarization. Our model also neglects any spin diffusion orthogonal to the sample growth $z$ direction. Furthermore, the nuclei of $^{75}$As, $^{69}$Ga, and $^{71}$Ga are not resolved in the present diffusion experiments. Therefore, these isotopes are treated as identical in the model since their dipole and quadrupole moments differ only by a factor of $\approx 2$ (see Supplementary Note 3). On the other hand, all these assumptions are justified since the very concept of classical spin diffusion is an inherently simplified description of the underlying quantum dynamics. As such, the diffusion coefficients $D$ should be treated as a coarse-grained description, aggregating the numerous lattice constant-scale parameters of the many-body spin ensemble evolution.

## Discussion

The GaAs/AlGaAs QDs grown by nanohole infilling combine excellent optical properties with low intrinsic strain, allowing for nuclear spin qubit and quantum memory designs[32,33,46]. The key performance characteristic is the nuclear spin coherence time, which can be extended up to $T_{2,N} \approx 10$ ms (ref. 33), but is ultimately limited by the longitudinal relaxation time $T_{1,N}$. Moreover, it is the state longevity of the nuclei interfaced with the QD electron spin that is relevant. Thus, one should consider the NSR time in the regime of short pumping, found here to range from $T_{1,N}^{(1e)} \approx 1$ s at low magnetic fields to $T_{1,N}^{(1e)} \approx 10$ s at high fields. For nuclear spin quantum computing with the typical 10

$\mu$s coherent control gates[33], a large number of operations $\geq 10^5$ would be possible without the disruptive effect of spin diffusion. Spatially inhomogeneous nuclear spin polarization, such as generated in the two-pump diffusion reflux experiment, may itself be of use for all-electrical control of the electron spin[78].

In conclusion, we have addressed the long-standing dilemma of whether the central spin of an electron accelerates or suppresses diffusion in a nuclear spin-lattice. We have used variable-duration optical pumping[15,17,20,22] to identify nuclear spin diffusion as the dominant NSR mechanism. In contrast to previous studies of nuclear spin diffusion[15,16,20,21,24,25], we use a charge tunable structure and probe nuclear spin dynamics with and without the electron under otherwise identical conditions – importantly, our QD charge control is achieved without reverting to optical pumping[24,25], thus eliminating the unwanted charge fluctuations. Combining these two aspects, we conclude that in a technologically important class of lattice-matched GaAs/AlGaAs nanostructures, the electron spin accelerates the nuclear spin diffusion, with no sign of a Knight-field-gradient barrier. We expect these findings to be relevant for a range of lattice-matched QDs[22,23,27,60] and shallow impurities[16], whereas an efficient spin diffusion barrier can arise from an electron with deep (sub-nanometer) localization[14]. Future work can examine the reduction of spin diffusion in low-strain nanostructures. The proximity of the $n$-doped layer, acting as a sink for nuclear polarization, as well as QD morphology, can be optimized. Alternatively, pure AlAs barriers can be used to grow GaAs QDs with well-isolated Ga nuclei, potentially offering long-lived spin memories and qubits.

## Data availability

The key data generated in this study are provided in the Source Data file SourceData.zip. The rest of the data that support the findings of this study are available from the corresponding author upon reasonable request. Source data are provided with this paper.

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

## Acknowledgements

P.M-H. and E.A.C. were supported by EPSRC through a doctoral training grant and award EP/V048333/1, respectively. E.A.C. was supported by a Royal Society University Research Fellowship. A.R.

acknowledges support of the Austrian Science Fund (FWF) via the Research Group FG5, I 4320, I 4380, I 3762, the European Union's Horizon 2020 research and innovation program under Grant Agreements No. 899814 (Qurope) and No. 871130 (Ascent+), the Linz Institute of Technology (LIT), and the LIT Secure and Correct Systems Lab, supported by the State of Upper Austria. E.A.C. is grateful to René Dost for advice on sample processing and would like to thank H.E. Dyte and G. Gillard for preliminary data on electron spin lifetimes.

## Author contributions

S.M., S.F.C.d.S., and A.R. developed, grew, and processed the quantum dot samples. P.M-H. and E.A.C. conducted the experiments and analysed the data. E.A.C. drafted the manuscript with input from all authors. E.A.C. performed numerical modeling and coordinated the project.

## Competing interests

The authors declare no competing interests.
