## [Peer Review File · Nature Communications]

REVIEWER COMMENTS

Reviewer #1 (Remarks to the Author):

This work brings some experimental novelty and analysis to the (now) rather old problem of nuclear spin diffusion in quantum dots. The problem of how nuclear polarization forms and then moves in III-V semiconductors has such a substantial qualitative literature it has become hard to track, but many outstanding questions in the microscopics persist. This work uses low-strain quantum dots, enabling reasonable RF excitation for NMR (unavailable in high-strain SK InGaAs dots), and of course the tool of optical pumping unavailable in gated dots, with optical neutral-exciton or trion-based read-out of Overhauser shifts. As such, a systematic study is possible of timescales for nuclear diffusion as a function of magnetic field, pumping time, and free diffusion time, for both charged and neutral dots. From this a model of nuclear diffusion is made, some aspects of which counter or refine suggestions from the vast qualitative literature on the topic.

This is a problem of interest to a healthy number of people, especially those working with semiconductor spin qubits needing to understand more whether nuclear spins can be used as a resource or not. However, as with all quantum dot papers, the key concern is the degree of universality of the paper's conclusions --- is this result a set of observations relevant only to this single dot, this single dot type, or can/should we generalize? A significant disappointment of this paper is that nearly all data comes from a single dot, QD1. Although others (QD2 and QD3) and a hint of data from QD2 appears in Fig. 4b (and supplement Fig. 5), I have significant doubts as to whether the conclusions of this paper go beyond the particular quirks of QD1.

Beyond the generalizations of the present result to other semiconductor dot systems, another disappointment concerns the strength of the paper's conclusions. The abstract states definitely that the paper will "show experimentally" that the electron spin accelerates spin diffusion, and the introduction promises to "distinguish with high accuracy" the effects of the electron on spin diffusion, providing the "answer" to the "long-standing question" as to whether diffusion is governed by dipole-dipole dynamics or by electron fluctuations in these systems. However, after the results are presented, the language changes to "we *speculate* that the intrinsic electron spin flips, governed e.g. by the phonon relaxation and cotunneling coupling to the electron reservoir of the n-doped layer, contribute to acceleration of nuclear spin diffusion . . . ". The latter language is more correct; the data presented allows little more than speculation. The field-dependent data of Fig. 4 in particular has a lot going on. It is made clear why we might expect low-field electron influence, i.e. the electron-mediated couplings, but as the field is increased up to 10 T (!) a lot is going on. The data have multiple features, and the density of points (and lack of multiple samples) leave it unclear whether the scatter of points in Fig.~4a is due to noise or not. The authors are clear to point out that this data is different, qualitatively, than expectations/observations in other GaAs-based systems, but their explanation of the structure of this data is quite lacking.

How is the electron driving diffusion at high field? If spin-flips, it strikes me that a correlation of the data in Fig. 4 with electron T1 (quite measurable with techniques on-hand) should, considering fluctuation-dissipation arguments, provide more compelling evidence. If phonons, perhaps temperature dependence could be revealing, although the discussion around supplementary figure 5 suggests that the study was not sufficiently thorough to blame or exonerate phonon processes for driving diffusion, and if phonons are to blame, whether it is dominated through spin-flip processes (when the dot is charged) or by "two-phonon quadrupolar relaxation", relevant when the dot is neutral (as in Supp. Fig. 5). If the relevant process for a charged dot is cotunneling, then shouldn't the diffusion rates depend on the diode bias? Where are these studies?

Surprisingly lacking is any analysis as to whether the exciton/trion measurement process contributes to driven diffusion. When the dot is empty, the optical driving of excitonic states for spectroscopically determining the Overhauser shift "suddenly" introduces an electron-spin, with significant "spare energy" to drive many possible dynamic hyperfine processes. When the dot is occupied, the driving of trions effectively and suddenly "removes" the electron spin, by pairing it into a singlet (and the hole has a very different hyperfine shift.) These sudden hyperfine shifts, (sometimes called "hyperfine shock"), have been the topic of similar study in phosphorus-in-silicon recently (cf. Morello and Simmons groups at UNSW), where there is perhaps more control on electron occupation and timing thereof (due to having single-shot, single-electron read-out) than in optically driven dots. There is significant reason to believe that the "spin flips" driving diffusion occur during the measurement as opposed to being due to phonon or cotunneling processes directly.

So in the bottom line, my disappointment with this paper is that "quantitative answering" of the question posed is lacking -- instead we see a qualitative analysis of a rather complicated (noisy?) dataset (Fig. 4) which leads to a qualitative speculation that driving of diffusion is happening due to an unidentified (or at least not distinguished) mechanism. As such, I disagree that the "answers" provided are useful to those seeking to design or use (or avoid) nuclear memory in GaAs systems. To warrant publication in Nature Communications, I would expect at least one of:

(1) Data such as Fig. 4a from more than one device. Quantum dots are especially notorious for portraying non-reproducible behavior. If the structure of Fig. 4 is real, then another device should show it. Something more rich is going on with magnetic field here, and without some understanding of it, I am not convinced the authors have measured what they claim. If another device shows very different Γ_N vs. B_z data, then the results are even more suspect.

(2) Studies that identify and validate a mechanism for the abstract's claim of "showing" that electron spins "accelerate nuclear spin diffusion." (The authors may argue that Fig. 4b "shows" this, regardless of whether they can explain it, but I don't believe they have successfully distinguished accelerated diffusion relative to other relaxation processes.)

If the authors have such data or can offer a model for the detailed dependence of Fig. 4a (or even have a denser dataset), I might be convinced otherwise. But without these criteria, this paper remains quite speculative. In any case, this is the flavor of result, with supplement included, I would expect to see in the pages of Physical Review B, as opposed to Nature Communications, in its present form.

Reviewer #2 (Remarks to the Author):

Millington-Hotze et al. experimentally investigated the physics of nuclear spin diffusion using a GaAs/AlGaAs quantum dot. Specifically, they explored the impact of a single electron spin inside the quantum dot on the nuclear spin diffusion rate. According to theories, the electron spin can create barriers for nuclear spin diffusion since it creates a Knight field that inhomogeneously shifts the frequency of the nuclear spins, but at the same time the electron spin may accelerate the nuclear spin diffusion via electron spin mediated nuclear spin flip-flop. It remains unclear which effect is dominant in the quantum dot system. The authors designed a number of experiments that convincingly show that the electron spin does accelerate the nuclear spin diffusion. Crucially, by using a quantum dot embedded inside a p-i-n diode, they can deterministically prepare the quantum dot in either a single-electron state or a neutral state, and directly compare the nuclear spin diffusion rate with and without a single electron spin. The results not only reveal interesting many-body physics, but may also provide useful insights in the future development of long-lived quantum memories using the nuclear spin ensembles inside a quantum dot.

Overall, I find this paper clear and well written, and the results are very interesting and impactful. I would support the publication of this paper in Nature Communications if the authors can address the following technical comments. Most of my technical comments are related with the details of the measurement techniques, which in my opinion were not described with enough details.

1. The paper lacks details in the measurement techniques, despite that there is a reasonably well-organized supplement. For example, for the pulse sequence shown at Fig. 2e and Fig. 3a, what is the duration and frequency of each pulse? It is useful to label the duration of each pulse and the delay between the different pulses in a duplicated figure placed in the supplement.
2. The authors use the spectral splitting of the two trion transitions of a negatively charged quantum dot to probe the nuclear spin polarization. An important question is how much backaction this measurement creates on the nuclear spin polarization. In the supplement, the authors did mention

that the nuclear spins are depolarized by a few percent due to the probe. But how was this few percent measured or estimated?

3. The authors generated the initial nuclear spin polarization by exciting the quantum dot with a circularly polarized pump laser. What is the physics of this process? The authors did not provide references, neither did they explain it in the supplement. Please add the proper explanation so that the readers can understand the experiment better.

4. Was As-75 the only nuclear spin species being studied in this experiment? If so, why As-75, instead of Ga-69 and Ga-71? Is it possible to extract additional information by studying the nuclear spin diffusion of the Ga-69 and Ga-71?

5. The physics process of the nuclear spin polarization erasure is not clear, despite that Ref. 46 is cited. Given that this is an important step of the experiment, the authors should provide some explanation in the supplement.

6. In Fig. 3b, it seems to me that the blue curve (with the electron) does not fit well to the measured data compared with the black one (without the electron). Could the authors comment on this?

7. In Fig. 4b, the data for a second quantum dot is shown, but only at $B = 0.5$ T. What is the purpose here? Have the authors measured the same B dependent NSR rate ratio for another dot?

8. Given that electron spin flips were considered as the reason for the unexpected large nuclear spin diffusion in the studied quantum dot, the authors should measure the electron spin lifetime of this dot. As far as I know, the electron spin lifetime can vary a lot between different wafers and even different dots on the same wafer.

In addition to the main comments described above, I also have a few minor comments listed below.

1. In both Fig. 2e inset and Fig. 3a, there is a symbol (1e) labeled on top of the V_{gate} curve, making it difficult to see the actual level of V_{gate} during the probe. I suggest moving this symbol away from the actual line.

2. The Hamiltonian shown in Eq. (1) contains no plank constant. However, when reporting the values of A_j , the authors still report the value of A_j/h (for example, in page 5, third paragraph). This is inconsistent.

3. There are two plus signs adjacent to each other in Eq. (2).

We would like to thank both Reviewers for their careful examination of our work and insightful suggestions. Based on these comments we have conducted new experiments and revised the manuscript thoroughly (changes highlighted in red). Below we give a point-by-point response – we believe we fully address all the criticism brought by the Reviewers. In what follows the original comments are highlighted in italics.

Reviewer #1 (Remarks to the Author):

This work brings some experimental novelty and analysis to the (now) rather old problem of nuclear spin diffusion in quantum dots. The problem of how nuclear polarization forms and then moves in III-V semiconductors has such a substantial qualitative literature it has become hard to track, but many outstanding questions in the microscopics persist. This work uses low-strain quantum dots, enabling reasonable RF excitation for NMR (unavailable in high-strain SK InGaAs dots), and of course the tool of optical pumping unavailable in gated dots, with optical neutral-exciton or trion-based read-out of Overhauser shifts. As such, a systematic study is possible of timescales for nuclear diffusion as a function of magnetic field, pumping time, and free diffusion time, for both charged and neutral dots. From this a model of nuclear diffusion is made, some aspects of which counter or refine suggestions from the vast qualitative literature on the topic.

This is a problem of interest to a healthy number of people, especially those working with semiconductor spin qubits needing to understand more whether nuclear spins can be used as a resource or not. However, as with all quantum dot papers, the key concern is the degree of universality of the paper's conclusions --- is this result a set of observations relevant only to this single dot, this single dot type, or can/should we generalize? A significant disappointment of this paper is that nearly all data comes from a single dot, QD1. Although others (QD2 and QD3) and a hint of data from QD2 appears in Fig. 4b (and supplement Fig. 5), I have significant doubts as to whether the conclusions of this paper go beyond the particular quirks of QD1.

We are pleased that the Reviewer recognizes that the subject addressed in our manuscript is of interest and thank them for appreciating the experimental novelty of our work. In response to Reviewer's concerns, we have conducted extensive additional experiments. For the revised paper we have collected data on a set of 5 randomly chosen QDs. As we describe below in more detail, the key results on nuclear spin diffusion are consistent in all studied dots.

*Beyond the generalizations of the present result to other semiconductor dot systems, another disappointment concerns the strength of the paper's conclusions. The abstract states definitely that the paper will "show experimentally" that the electron spin accelerates spin diffusion, and the introduction promises to "distinguish with high accuracy" the effects of the electron on spin diffusion, providing the "answer" to the "long-standing question" as to whether diffusion is governed by dipole-dipole dynamics or by electron fluctuations in these systems. However, after the results are presented, the language changes to "we *speculate* that the intrinsic electron spin flips, governed e.g. by the phonon relaxation and cotunneling coupling to the electron reservoir of the n-doped layer, contribute to acceleration of nuclear spin diffusion . . . ". The latter language is more correct; the data presented allows little more than speculation.*

There is a hierarchy of conclusions that we make in this work. The key points are as follows:

1. The central spin accelerates nuclear spin relaxation. This is an experimental fact which doesn't need any speculation and essentially follows from the raw data.
2. A direct conclusion from the above is the disproval of the long-circulated concept of a Knight-field-gradient diffusion barrier – we conclude that it doesn't apply to the studied epitaxial GaAs quantum dots for a very wide range of magnetic fields.
3. Our next conclusion is that electron spin actually accelerates spin diffusion. This conclusion requires a bit more logical reasoning, but is still rather straightforward. Here we use variable duration spin pumping to distinguish diffusion from all other known relaxation mechanisms (long pumping essentially switches off diffusion). Moreover, in order to close any potential loopholes, we have conducted new experiments to get by far the most direct proof that spin diffusion is indeed the dominant relaxation mechanism – this is discussed in more detail below.
4. Finally, the aforementioned acceleration of spin diffusion is observed up to unexpectedly large magnetic fields. Here, we propose an explanation based on the role of the electron spin flip fluctuations. This is the only major conclusion which indeed can be described as speculation/conjecture. Even so, this conjecture is based on the relevant theoretical work [e.g. Horvitz, PRB 3, 2868 (1971)]. Our explanation is also corroborated by our data – a numerical model fitting shows that optical excitation (which inevitably causes electron spin to fluctuate) accelerates spin diffusion. Future work may provide a more direct experimental evidence – in the revised version we briefly outline how this can be approached.

Following the Reviewer's comment, we have modified the abstract to make a clear distinction between our observations, conclusions, and conjectures (which are all important elements in research). As outlined above, most of our report is based on direct experimental observations and rigorous logical conclusions. Accelerated spin diffusion specifically at high magnetic fields is the only part where our knowledge is currently limited to speculation / proposed explanation.

The field-dependent data of Fig. 4 in particular has a lot going on. It is made clear why we might expect low-field electron influence, i.e. the electron-mediated couplings, but as the field is increased up to 10 T (!) a lot is going on. The data have multiple features, and the density of points (and lack of multiple samples) leave it unclear whether the scatter of points in Fig.~4a is due to noise or not. The authors are clear to point out that this data is different, qualitatively, than expectations/observations in other GaAs-based systems, but their explanation of the structure of this data is quite lacking.

Based on Reviewer's comment we noted that the data was not well presented in the original version. We have changed the vertical scale of the plot (now Fig. 5a) to match the plots of the preceding figure (now Figs. 4 c,d). Comparing the two figures,

one can see that in fact not much is going on when magnetic field is varied. Instead the relaxation rate is dominated by the pumping time T_{Pump} (because diffusion is the dominant NSR mechanism) – NSR rates continue to depend on T_{Pump} without saturation even for $T_{\text{Pump}} \sim 1000\text{s}$ (Figs. 4c,d). But even when T_{Pump} is fixed, the actual nuclear spin pumping conditions inevitably change with magnetic field, for example because the absorption spectrum of the quantum dot / quantum well is field dependent, or because laser focusing changes slightly. We attribute the residual scatter of the NSR rates measured at different magnetic fields in QD1 (Fig. 5a) to such fluctuations. Moreover, we have collected new data for another four QDs (Fig. 5b) at selected magnetic fields, and we observe similar random scatter. We have modified the text to provide this explanation, stressing that the direct dependence of NSR on magnetic field is a weak effect.

**How* is the electron driving diffusion at high field? If spin-flips, it strikes me that a correlation of the data in Fig. 4 with electron T_1 (quite measurable with techniques on-hand) should, considering fluctuation-dissipation arguments, provide more compelling evidence. If phonons, perhaps temperature dependence could be revealing, although the discussion around supplementary figure 5 suggests that the study was not sufficiently thorough to blame or exonerate phonon processes for driving diffusion, and if phonons are to blame, whether it is dominated through spin-flip processes (when the dot is charged) or by "two-phonon quadrupolar relaxation", relevant when the dot is neutral (as in Supp. Fig. 5). If the relevant process for a charged dot is cotunneling, then shouldn't the diffusion rates depend on the diode bias? Where are these studies?*

Firstly, we need to make a distinction between the different effects that phonons and cotunneling can induce. On the one hand, there are nondiffusion mechanisms, where phonons and the electron Fermi reservoir act as the ultimate sink for the nuclear spin momentum, which is carried through the electron spin. On the other hand, we propose here that electron spin fluctuations can mediate nuclear-nuclear flip-flops, which then accelerate the nuclear spin diffusion. The processes are distinct, because diffusion is spin-conserving, whereas non-diffusion relaxation mechanisms involve transfer of spin momentum from nuclei to other degrees of freedom. The manuscript has been revised accordingly – when we present our Knight-field-fluctuation hypothesis, we stress that it's distinct from the non-diffusion mechanisms. Conversely, whenever pure relaxation mechanisms are discussed, we further stress their non-diffusion nature.

We have conducted preliminary electron spin lifetime (T_{1e}) measurements in our quantum dot sample. There is a wealth of new techniques, data and physics, which will be published separately. In the revised manuscript we only quote the T_{1e} values relevant to the subject of the present work (nuclear spin diffusion). Relaxation accelerates with magnetic field, changing from $T_{1e} = 7\text{ ms}$ at 2 T to $T_{1e} = 0.5\text{ ms}$ at 7 T. The electron flips occur as abrupt jumps (telegraph process), so the Knight field should have significant spectral density not only at $1/T_{1e}$, but at lower and

higher frequencies as well. Therefore, the fluctuating electron spin should provide frequency components around the [1,10] kHz range, matching the typical differences in the nuclear spin energies as revealed by NMR spectra. Under these conditions the mechanism described by Horvitz [PRB 3, 2868 (1971)] becomes applicable. For example, if two nuclei (nearby or distant) have a frequency mismatch of 3 kHz, it is too large for the dipolar reservoir to provide or absorb such energy. But if the nuclei are subject to a 3 kHz oscillating field (arising from electron spin fluctuations), this can effectively “bridge the energy gap”, making nuclear flip-flop possible, thus accelerating spin diffusion. Again, in this diffusion mechanism, the phonons are only causing the electron spin to fluctuate. This should not be confused with the non-diffusion phonon-related relaxation channels.

Specifically for the two-phonon quadrupolar (non-diffusion) relaxation, which does not involve electron spin, our temperature-dependent experiments (now Supplementary Fig. 7) show that such contribution is small. As we clarify in the revised version, this quadrupolar relaxation is expected to be small at $T < 20$ K, based on the previous bulk crystal studies [PRB 13, 4705 (1976)].

Bias-dependent NSR experiments have been conducted, with an example shown in Supplementary Fig. 5. The increase in cotunneling indeed accelerates NSR. But as we clarify in the revised version, the spin momentum of the nuclei is simply carried into the Fermi reservoir by the electron cotunneling events – it’s a non-diffusion mechanism. Consequently, not much can be deduced about nuclear spin diffusion from the bias dependent measurements.

Surprisingly lacking is any analysis as to whether the exciton/trion measurement process contributes to driven diffusion. When the dot is empty, the optical driving of excitonic states for spectroscopically determining the Overhauser shift "suddenly" introduces an electron-spin, with significant "spare energy" to drive many possible dynamic hyperfine processes. When the dot is occupied, the driving of trions effectively and suddenly "removes" the electron spin, by pairing it into a singlet (and the hole has a very different hyperfine shift.) These sudden hyperfine shifts, (sometimes called "hyperfine shock"), have been the topic of similar study in phosphorus-in-silicon recently (cf. Morello and Simmons groups at UNSW), where there is perhaps more control on electron occupation and timing thereof (due to having single-shot, single-electron read-out) than in optically driven dots. There is significant reason to believe that the "spin flips" driving diffusion occur during the measurement as opposed to being due to phonon or cotunneling processes directly.

Firstly, we note that optical probing (measurement) stage is separated in time from the spin diffusion, which always occurs in the dark in our experiments. Thus, the probe cannot affect the nuclear spin diffusion dynamics directly. An indirect influence is possible, in a sense that the optical probe pulse induces nuclear depolarization and can in principle distort the measured value of the post-diffusion nuclear spin polarization. In order to avoid such distortions, we calibrate the duration of the optical probe pulse whenever magnetic field is changed considerably. To clarify this point, we show the results of a typical calibration experiment in

Supplementary Fig. 3. The duration of the probe pulse was varied over >3 orders of magnitude, and the probe's effect on the observed nuclear spin polarization was measured. In the example shown we have chosen a probe time of 75 ms. It is short enough to have negligible effect on the measured nuclear hyperfine shift, but is sufficiently long to collect enough QD photoluminescence photons.

The lack of any “hyperfine shock” can be explained by the spin conservation and the different number of nuclear spins in GaAs QDs and Si dopants. In a typical QD there are 10^5 spin-3/2 nuclei, so it would take on the order of 10^5 electron-nuclear spin flip-flops to bring enough spin momentum to change the average nuclear spin polarization of the QD. Moreover, only a small fraction of the optically injected electrons exchange spin with the nuclei (due to the 3 orders of magnitude mismatch between electron and nuclear Zeeman energies). This explains why optical probing results in a smooth and relatively slow (few seconds) degradation of the average nuclear spin polarization. In the revised manuscript this point is answered with the addition of Supplementary Fig. 3 and the relevant description.

So in the bottom line, my disappointment with this paper is that "quantitative answering" of the question posed is lacking -- instead we see a qualitative analysis of a rather complicated (noisy?) dataset (Fig. 4) which leads to a qualitative speculation that driving of diffusion is happening due to an unidentified (or at least not distinguished) mechanism. As such, I disagree that the "answers" provided are useful to those seeking to design or use (or avoid) nuclear memory in GaAs systems.

This is related to the previous points – as explained above, we now make a clear distinction between robust conclusions and conjectures. Summarising on the detailed answers to the specific points above, we argue that the mechanism driving nuclear spin diffusion has been identified in our experiments. It is the nuclear-nuclear spin flip-flop mediated by the electron central spin. This mechanism is known. However, the bulk of the previous work assumed the electron to be static. This assumption can explain accelerated spin diffusion at low magnetic fields, but fails to account for what we observe at high fields. We argue that this can be rectified by considering the dynamical electron spin fluctuations. In other words, there is a well-known model which considers only the zero-frequency of the electronic spin evolution spectrum. We conjecture that the entire field dependence can be explained by considering the entire electronic spin spectrum, in line with the model of Horvitz [PRB 3, 2868 (1971)]. We believe this provides a conclusive answer about the dominant role of the electron in nuclear spin diffusion, and identifies the direction for future work on the role of electron spin flips.

To warrant publication in Nature Communications, I would expect at least one of: (1) Data such as Fig. 4a from more than one device. Quantum dots are especially notorious for portraying non-reproducible behavior. If the structure of Fig. 4 is real, then another device should show it. Something more rich is going on with magnetic field here, and without some

understanding of it, I am not convinced the authors have measured what they claim. If another device shows very different Γ_N vs. B_z data, then the results are even more suspect.

For the revised manuscript we have conducted additional experiments and measured nuclear spin relaxation in several individual quantum dots. Due to the long nuclear spin relaxation times the experiments are inherently slow – for any process where relaxation happens on a timescale of a few minutes, it takes many hours to conduct a point-by-point collection of a detailed dynamics trace. Thus, it was not possible to collect full magnetic field dependence of the NSR rates in multiple dots. We've circumvented this difficulty by measuring NSR rate ratios in two extreme cases – at low magnetic field (0.5 T) and at high magnetic field (10 T). These results are now shown in Fig. 5d. Overall, we have data for 5 dots, of which 4 were measured at high field (one dot wasn't measured at high field to save time).

That data confirms the significance of the conclusion derived from the full magnetic field dependence obtained for QD1. In particular, we find that:

- In all studied dots the central electron spin accelerates NSR, confirming our conclusion that no significant Knight-field-gradient diffusion barrier is formed in quantum dots.
- At low magnetic field, in all studied dots we find that NSR acceleration is disproportionately stronger at short pumping. The actual ratios vary between individual dots, for the same reason why the absolute nuclear spin decay rates vary with magnetic field (variation of the effective spin pumping rate), as already explained above. However, the excess of the NSR $1e/0e$ rate ratio at short pumping is a robust effect, which supports our conclusion about acceleration of spin diffusion by the central spin.
- At high magnetic field, the difference between short and long pumping diminishes, consistent with full magnetic field dependence on QD1 and our explanation based on the hyperfine-mediated nuclear-nuclear interactions (Eq.3).

Thus, we show that all studied individual quantum dots demonstrate the same type of behaviour, further strengthening our conclusions.

(2) Studies that identify and validate a mechanism for the abstract's claim of "showing" that electron spins "accelerate nuclear spin diffusion." (The authors may argue that Fig. 4b "shows" this, regardless of whether they can explain it, but I don't believe they have successfully distinguished accelerated diffusion relative to other relaxation processes.)

The first step in addressing this question is to assert the existence of spin diffusion. If one could monitor the state of each individual nucleus, that could provide a direct evidence of spin diffusion. But this is well beyond reach for any experimental implementation. Thus, we need to find signatures that would distinguish diffusion from other mechanism.

For the revised manuscript we've developed the following approach. For any non-diffusion dissipation mechanism the dynamics of nuclear spin polarization is expected to be monotonic in time. And this is what is usually observed – an exponential decay of the average nuclear polarization towards its equilibrium level. By contrast, the diffusion equation intertwines the spatial and temporal variables. We use this feature to engineer non-monotonic (oscillatory) free relaxation towards equilibrium. This is achieved by time-oscillating nuclear spin pumping, which, through diffusion, creates a spatial polarization wave. The pumping is then switched off and polarization relaxation is monitored at a certain point in space (QD position). Diffusion can then lead to oscillatory free relaxation (i.e. in the absence of external pumping). Experimentally we use a sequence of two optical pumps that polarize the nuclei first in the negative and then in the positive direction with respect to the static external magnetic field. We then monitor the dynamics of the nuclear polarization at the quantum dot site and indeed observe oscillatory relaxation (new Fig. 3). This situation can be described as “diffusion reflux”: the negative polarization generated by the first pump is stored in the volume outside the dot but then diffuses back (in the dark) into the dot, overwhelming its positive polarization that was induced by the second pump. This is similar to “hole burning” experiments on GaAs shallow donors [PRB 25, 4444 (1982)] To our knowledge, for quantum dots this is by far the most direct demonstration of diffusive nuclear spin dynamics.

Once the existence of diffusion is established, the next logical step follows. If a quantum dot is pumped for a long time, diffusion establishes a more uniform spatial distribution of the nuclear spin polarization. That uniformity slows down the subsequent free diffusion, after the pump is switched off. This gives us a reliable tool: by varying the pumping time we can distinguish diffusion and non-diffusion relaxation mechanisms. Furthermore, following Reviewer's comments we have verified these results on multiple individual quantum dots. With the addition of the new experimental data, we believe we have a strong argument to support our conclusion that electron spin accelerates spin diffusion.

If the authors have such data or can offer a model for the detailed dependence of Fig. 4a (or even have a denser dataset), I might be convinced otherwise. But without these criteria, this paper remains quite speculative. In any case, this is the flavor of result, with supplement included, I would expect to see in the pages of Physical Review B, as opposed to Nature Communications, in its present form.

As outlined above, we've accomplished not just one of, but both items recommended by the Reviewer. The magnetic field dependence (now Fig. 5a) is presented in a better way to demonstrate that pumping time is the dominant factor, whereas the magnetic field dependence is in fact rather weak.

While spin diffusion has been studied previously over many decades, our present work is by far the most systematic study for semiconductor nanostructures, featuring independent charge control, radiofrequency manipulation of nuclear spins and a wide range of magnetic fields. Thanks to this systematic approach we are finally able to resolve the dilemma of what an electron does to nuclear spin diffusion. Our results show that for quantum dots there is no Knight-field-gradient barrier – the opposite effect of diffusion acceleration prevails for all realistic conditions. Beyond strengthening the core message of the paper, the new diffusion reflux experiment is by itself a neat result – to our knowledge this is the most direct (assumption-free) visualization of nuclear spin diffusion in an individual quantum dot. We hope these arguments convince the Reviewer that our work is a significant step forward in the study of spin diffusion.

Reviewer #2 (Remarks to the Author):

Millington-Hotze et al. experimentally investigated the physics of nuclear spin diffusion using a GaAs/AlGaAs quantum dot. Specifically, they explored the impact of a single electron spin inside the quantum dot on the nuclear spin diffusion rate. According to theories, the electron spin can create barriers for nuclear spin diffusion since it creates a Knight field that inhomogeneously shifts the frequency of the nuclear spins, but at the same time the electron spin may accelerate the nuclear spin diffusion via electron spin mediated nuclear spin flip-flop. It remains unclear which effect is dominant in the quantum dot system. The authors designed a number of experiments that convincingly show that the electron spin does accelerate the nuclear spin diffusion. Crucially, by using a quantum dot embedded inside a p-i-n diode, they can deterministically prepare the quantum dot in either a single-electron state or a neutral state, and directly compare the nuclear spin diffusion rate with and without a single electron spin. The results not only reveal interesting many-body physics, but may also provide useful insights in the future development of long-lived quantum memories using the nuclear spin ensembles inside a quantum dot. Overall, I find this paper clear and well written, and the results are very interesting and impactful. I would support the publication of this paper in Nature Communications if the authors can address the following technical comments. Most of my technical comments are related with the details of the measurement techniques, which in my opinion were not described with enough details.

We thank the Reviewer for the positive assessment of our work and respond to the comments below.

1. The paper lacks details in the measurement techniques, despite that there is a reasonably well-organized supplement. For example, for the pulse sequence shown at Fig. 2e and Fig. 3a, what is the duration and frequency of each pulse? It is useful to label the duration of each pulse and the delay between the different pulses in a duplicated figure placed in the supplement.

We have added further information on radiofrequencies and a more detailed timing diagram in a new Supplementary Figure 2. All pulse durations and interpulse delays are labelled and discussed in Supplementary Note 2 – this discussion has been extended to provide all the relevant numerical values.

2. The authors use the spectral splitting of the two trion transitions of a negatively charged quantum dot to probe the nuclear spin polarization. An important question is how much backaction this measurement creates on the nuclear spin polarization. In the supplement, the authors did mention that the nuclear spins are depolarized by a few percent due to the probe. But how was this few percent measured or estimated?

The answer is related to a similar question from Reviewer 1. To clarify this point, we show the results of a typical calibration experiment in Supplementary Fig. 3. The duration of the probe pulse was varied over >3 orders of magnitude, and the probe's effect on the observed nuclear spin polarization was measured. In this example we have chosen a probe time of 75 ms, which can be seen to have negligible effect on the measured nuclear hyperfine shift. Such calibration was conducted for each individual quantum dot and repeated when magnetic field was changed considerably. This way we ensure that the probe-induced error does not exceed a few percent.

3. The authors generated the initial nuclear spin polarization by exciting the quantum dot with a circularly polarized pump laser. What is the physics of this process? The authors did not provide references, neither did they explain it in the supplement. Please add the proper explanation so that the readers can understand the experiment better.

We have added a brief description of the nuclear spin pumping process. Since it's a well-studied subject, we present this information in Supplementary (Note 2), citing several original sources and one review paper. Relevant references are also given in the main text. In a nutshell, nuclear spin pumping is a cyclic process. The role of the circularly polarized laser is to generate spin-polarized electrons using the selection rules of the optical transitions between heavy-hole and conduction bands in group III-V semiconductors. The polarized electron then passes its spin to one of the nuclear spins via the flip-flop part of the hyperfine (magnetic) interaction. The cycle is completed by optical recombination, which removes the electron. By repeatedly exciting the quantum dot with a laser, it is then possible to polarize a significant fraction of the $\sim 10^5$ nuclei in a typical quantum dot.

4. Was As-75 the only nuclear spin species being studied in this experiment? If so, why As-75, instead of Ga-69 and Ga-71? Is it possible to extract additional information by studying the nuclear spin diffusion of the Ga-69 and Ga-71?

Our experiments on spin diffusion were done without resolving the individual isotopes. This is now clarified early on in the text, where we state that the key measurable quantity (the Overhauser shift) is an unresolved sum from all isotopes with contributions scaling approximately as 49:28:23 % for ^{75}As : ^{69}Ga : ^{71}Ga . Fundamentally, there should be no obstacle in measuring diffusion of individual isotopes, but that would come at a price of a considerable extra complexity in

experimental implementation. At present, we do not think it is worth the effort for the following reason. For any given quantum dot shape and size, all the difference in spin diffusion would be governed by the three isotope-specific parameters: the magnetic dipole moment, the electric quadrupolar moment and the isotope abundance. However, the difference in these parameters is only a factor of $\sim 2 - 3$ for the three abundant isotopes of As and Ga. First principle estimates of Supplementary Note 3 show that the largest difference in diffusion coefficients (between ^{75}As and ^{71}Ga) is less than a factor of 3. We now highlight these scales of isotopic difference in the Discussion part of the paper. Such differences can affect the exact nuclear spin relaxation rates, but are unlikely to result in fundamentally different physics. Previous isotope-dependent studies, for example on nuclear spin coherence [Nature Comm 13, 4048], support this reasoning. One exception is the qualitatively different behaviour of the spin-9/2 indium coherence compared to the spin-3/2 isotopes of As and Ga [Nature Comm 10, 3157]. However, in the GaAs/AlGaAs QDs studied here, all the isotopes are spin-3/2 and we expect the same qualitative behaviour.

The NMR spectra in the manuscript are indeed shown for ^{75}As only. However, the effect of strain on different isotopes is known from prior studies. In particular, the nuclear quadrupolar shifts of ^{69}Ga and ^{71}Ga are approximately 2 and 3 times smaller than for ^{75}As , when subject to the same strain. We have highlighted this difference in the revised manuscript.

5. The physics process of the nuclear spin polarization erasure is not clear, despite that Ref. 46 is cited. Given that this is an important step of the experiment, the authors should provide some explanation in the supplement.

We have added a clarification in Supplementary Note 2 to explain the essence of nuclear magnetic resonance saturation. When subject to an oscillating magnetic field, resonant with the nuclear Larmor frequency, the nuclear spins undergo Rabi rotation, periodically transitioning between the spin states parallel and antiparallel to the external magnetic field. Due to the nuclear-nuclear dipolar interactions each nuclear spin is subject to a local field. The randomness of these local fields perturbs the Rabi precession frequencies, resulting in ensemble dephasing. Consequently, the nuclei become randomly oriented (depolarized) after a long resonant radiofrequency saturation pulse.

6. In Fig. 3b, it seems to me that the blue curve (with the electron) does not fit well to the measured data compared with the black one (without the electron). Could the authors comment on this?

In the last paragraph of the Results section we list a number of simplifying assumptions built into our diffusion equation model (for example the isotope difference in the spin dynamics discussed in the context of point 4 above, or the assumption about the one-dimensional character of the diffusion process). In the

revised manuscript we point out explicitly that such simplifications are likely to be responsible for the imperfect fitting, which is indeed apparent in Fig. 4b both for the 0e and 1e curves.

One may ask if the fitting can be improved with a more complex diffusion model, for example by introducing isotope-specific diffusion coefficients and extra parameters to account for the anisotropy of diffusion in the sample growth plane and along the growth axis? Of course, technically the fitting would be improved, simply through a larger number of free parameters. But we don't believe that it would give any genuine insights. Fundamentally, the (classical) spin diffusion description is already a simplification of a quantum many-body problem, so we think it would be unnecessary to push the diffusion model beyond its validity range. This is why we present the diffusion model fitting in Fig. 4b as it is, to demonstrate both that it captures well the main features of the spin dynamics, and that it inevitably fails in describing the details. We have highlighted these points in the last paragraph of the Discussion section.

7. In Fig. 4b, the data for a second quantum dot is shown, but only at $B = 0.5$ T. What is the purpose here? Have the authors measured the same B dependent NSR rate ratio for another dot?

The answer is related to a similar question from Reviewer 1. For the revised manuscript we have conducted additional experiments and measured nuclear spin relaxation in several individual quantum dots. Due to the long nuclear spin relaxation times the experiments are inherently slow – for any process where relaxation happens on a timescale of a few minutes, it takes many hours to conduct a point-by-point collection of a detailed dynamics trace. Thus, it was not possible to collect full magnetic field dependence of the NSR rates in multiple dots. We've circumvented this difficulty by measuring NSR rate ratios in two extreme cases – at low magnetic field (0.5 T) and at high magnetic field (10 T). These results are now shown in Fig. 5c. Overall, we have data for 5 dots, of which 4 were measured at high field. The data confirms the significance of the conclusion derived from the full magnetic field dependence obtained for QD1, namely that the electron spin can only accelerate nuclear spin diffusion, and that no significant Knight-field-gradient barrier is formed in quantum dots.

8. Given that electron spin flips were considered as the reason for the unexpected large nuclear spin diffusion in the studied quantum dot, the authors should measure the electron spin lifetime of this dot. As far as I know, the electron spin lifetime can vary a lot between different wafers and even different dots on the same wafer.

The answer is related to a similar question from Reviewer 1. We have conducted preliminary measurements of the electron spin lifetime (T_{1e}) in our quantum dot sample. The details will be published separately. In the revised manuscript we only quote the T_{1e} values relevant to the subject of the present work (nuclear spin diffusion). We find that the main relaxation channel of the electron spin is due to the phonon-related processes – this is now mentioned in the manuscript. In the phonon-limited case, the lifetimes are very similar in individual QDs, and even in different structures. A good example is InAs QDs, where several independent studies have produced very similar numbers. For example, at $B=6$ T and $T=4.2$ K the phonon-limited spin relaxation time is 200 μ s as shown both in [PRB 81 035332 (2010)] and [npj Quantum information 7, 43 (2021)]. Our ongoing experiments yield the same conclusion for GaAs/AlGaAs quantum dots – the electron spin lifetimes are indeed similar in different dots from the same sample.

In addition to the main comments described above, I also have a few minor comments listed below.

1. In both Fig. 2e inset and Fig. 3a, there is a symbol ($1e$) labeled on top of the V_{gate} curve, making it difficult to see the actual level of V_{gate} during the probe. I suggest moving this symbol away from the actual line.

The labels have been moved away from the lines.

2. The Hamiltonian shown in Eq. (1) contains no plank constant. However, when reporting the values of A_j , the authors still report the value of A_j/h (for example, in page 5, third paragraph). This is inconsistent.

The hyperfine constants A_j have the units of energy. This is now stated explicitly when the A_j symbol is defined in the manuscript. Thus, the Hamiltonian of Eq.1 has the units of energy, as it should. However, interpreting A_j in energy units, such as Joules or electronvolts is not convenient. It is more convenient to analyse the frequency shift (known as the Knight shift) that the electron spin imposes on the nuclear magnetic resonance. This frequency shift is $A_j/(2h)$, where the factor of $1/2$ is the electron spin projection, and the Planck constant is needed to convert the A_j energies into spectroscopic frequency shifts. In order to address any potential confusion, in the revised manuscript we state explicitly that A_j is an energy and $A_j/(2h)$ stands for a “Knight frequency shift”.

3. There are two plus signs adjacent to each other in Eq. (2).

The extra plus sign has been removed.

REVIEWERS' COMMENTS

Reviewer #1 (Remarks to the Author):

I have taken a long time in this second round of review, largely because I find this paper increasingly difficult to read. Its conclusions are a bit of a jumble. Even the authors' answers in their rebuttal admit that the paper requires a "hierarchy of conclusions". It takes me a tremendous amount of concentration to follow the threads of the argument, and other responsibilities continually interrupt. I just received a note that I am out of time, which is understandable given how much time I've spent.

I also kept putting it down since I find statements such as "A fluctuating electron spin can accelerate nuclear spin diffusion...– this contribution has been previously ignored in the context of III-V semiconductor nanostructures" to be irritating, as someone who has not ignored this mechanism and has even done calculations. [I do not need citations]. It is a messy topic, not always worthy of publication, but to suggest it has been "ignored" is unfair to many who have come before. Any claims to primacy should be earned on the basis of clarity and convincingness of the data, which remains marginal on this point.

Anyway, these are qualitative complaints about the style of writing of this work. I think the authors have done an admirable job of providing additional data and clearer figures to my first round issues, although clearer studies correlating T1 or T1rho of electron spins to nuclear diffusion/relaxation rates would really help make their conclusions seem less speculative; perhaps I must wait for the next paper for that, but I probably will not review it.

I do not think any of the data is **wrong** in this paper, and as per discussions about peer review, I believe that is where we should leave it, in the interest of time. It seems the phenomena are reasonably reproducible across dots, the model provided, to the extent that I can follow it, are consistent, and the conclusions are appropriately hedged when uncertain.

I think this paper will have trouble gaining broad attention, as it takes a hefty level of commitment to unpack it. However, this is an editorial decision, not a peer-review one. As far as I can tell, the data and conclusions are publishable.

Reviewer #2 (Remarks to the Author):

The authors have addressed nearly all my comments, and the revised manuscript has improved significantly. I have one remaining concern though. Given that the authors attribute the accelerated spin diffusion to electron spin flips, it is important to provide the measurement of the spin lifetime by directly showing the data. In the manuscript, the authors say that they did perform preliminary measurement and the data will be reported elsewhere, without a citation. This is a bit concerning since this data is an important piece supporting this conjecture about accelerated spin diffusion at large magnetic field. I would prefer the authors show their preliminary data of the spin lifetime measurement in the supplement.

RESPONSE TO REVIEWERS' COMMENTS

(Reviewers' comments are reproduced in *italics*.)

Reviewer #1 (Remarks to the Author):

I have taken a long time in this second round of review, largely because I find this paper increasingly difficult to read. Its conclusions are a bit of a jumble. Even the authors' answers in their rebuttal admit that the paper requires a "hierarchy of conclusions". It takes me a tremendous amount of concentration to follow the threads of the argument, and other responsibilities continually interrupt. I just received a note that I am out of time, which is understandable given how much time I've spent.

Following the Reviewer's comments, we have done a thorough proof-reading of the manuscript. Without changing the contents, we have restructured the text in many parts in order to improve readability and clarity. The changes are highlighted in red in the revised version.

On the other hand, it must be said that some hierarchy of conclusions and complexity is inevitable. Nuclear spin diffusion is a decades-old topic. Dozens of studies have been conducted previously, but with no agreement on a simple question – does the central electron spin accelerate or suppress nuclear spin diffusion? Each of the alternatives has been claimed to be the right one, creating some degree of confusion. Unravelling this confusion cannot be an easy task. Only by combining the correct type of sample structures, advanced experimental techniques, as well as high-sensitivity instrumentation, we were able to get a definitive experimental answer. Consequently, laying out these findings requires some hierarchy, and some patience from the reader. But we believe it is worth it – resolving simple questions with no simple answers is the very purpose (and the beauty) of the scientific method.

I also kept putting it down since I find statements such as "A fluctuating electron spin can accelerate nuclear spin diffusion...– this contribution has been previously ignored in the context of III-V semiconductor nanostructures" to be irritating, as someone who has not ignored this mechanism and has even done calculations. [I do not need citations]. It is a messy topic, not always worthy of publication, but to suggest it has been "ignored" is unfair to many who have come

before. Any claims to primacy should be earned on the basis of clarity and convincingness of the data, which remains marginal on this point.

We appreciate the Reviewer's comment and agree, that our initial wording could be due to us missing the relevant prior work. Unfortunately, despite a very thorough search of the relevant literature, we were unable to locate any detailed consideration of how electron spin fluctuations in III-V semiconductor nanostructures affect nuclear spin diffusion. The studies by Khutsishvili, Horvitz and Wolfe, conducted before the age of nanostructures, are the closest match we could find. On the other hand, our point is not about electron spin fluctuations being ignored historically. The key message is that they must be considered in III-V nanostructures. We have therefore reworded as follows: "This contribution has been considered for deep impurities, and, as we now discuss, should also be taken into account in the context of III-V semiconductor nanostructures."

Anyway, these are qualitative complaints about the style of writing of this work. I think the authors have done an admirable job of providing additional data and clearer figures to my first round issues, although clearer studies correlating T_1 or $T_1\rho$ of electron spins to nuclear diffusion/relaxation rates would really help make their conclusions seem less speculative; perhaps I must wait for the next paper for that, but I probably will not review it.

*I do not think any of the data is *wrong* in this paper, and as per discussions about peer review, I believe that is where we should leave it, in the interest of time. It seems the phenomena are reasonably reproducible across dots, the model provided, to the extent that I can follow it, are consistent, and the conclusions are appropriately hedged when uncertain.*

I think this paper will have trouble gaining broad attention, as it takes a hefty level of commitment to unpack it. However, this is an editorial decision, not a peer-review one. As far as I can tell, the data and conclusions are publishable.

We would like to thank the Reviewer for acknowledging the consistency and clarity of our additional data and figures. Moreover, in the revised manuscript we have added more data on electron spin relaxation times $T_{1,e}$. Although the data is preliminary (full study to be reported elsewhere), it provides sufficient information for the interpretation of spin diffusion. $T_{1,e}$ can be varied with magnetic field, but the field also has a direct effect on the electron-mediated nuclear spin diffusion. Temperature can be used to alter $T_{1,e}$, but the upper range is limited by the electron confinement energy. Proper correlation studies would require an independent

wide-range tuning knob for $T_{1,e}$, such as resonant microwave driving. This approach would not be without challenges either, for example due to the need to suppress the electron-nuclear double resonance. Hence, this is not something that can be easily done in a near future.

Our experimental study of nuclear spin diffusion focuses specifically on III-V semiconductor nanostructures. However, the importance and significance of nuclear spin diffusion is much broader. In the revised Introduction we cite a selection of applications, such as NMR signal enhancement and structural analysis of polymers, biomolecules, proteins and pharmaceutical formulations. While the specific parameters are different, the underlying physics, including the impact of the unpaired electrons on nuclear spin diffusion, are similar. Hence, we argue that our study will have a broader impact, beyond the realm of inorganic semiconductor devices.

Reviewer #2 (Remarks to the Author):

The authors have addressed nearly all my comments, and the revised manuscript has improved significantly. I have one remaining concern though. Given that the authors attribute the accelerated spin diffusion to electron spin flips, it is important to provide the measurement of the spin lifetime by directly showing the data. In the manuscript, the authors say that they did perform preliminary measurement and the data will be reported elsewhere, without a citation. This is a bit concerning since this data is an important piece supporting this conjecture about accelerated spin diffusion at large magnetic field. I would prefer the authors show their preliminary data of the spin lifetime measurement in the supplement.

Following the Reviewers suggestion, we have added examples of electron spin relaxation data in Supplementary Note 5, including Supplementary Figure 10. We describe the measurement methodology, which follows a recent work [Nature Communications 13, 4048 (2022)] with some modifications. This technique uses single-shot measurement of the electron spin via its nuclear spin environment in order to monitor the electron spin dynamics. In Supplementary Figures 10a-c we present examples of single-shot readout histograms, and then demonstrate in Supplementary Figure 10d how these histograms are used to derive the electron spin lifetime $T_{1,e}$. We quote electron $T_{1,e}$ values relevant to this work. A detailed study, including magnetic field dependence of $T_{1,e}$, is more appropriate for a separate publication due to the volume of data and the difference in the scope.